# TNFR1-dependent cell death drives inflammation in *Sharpin*-deficient mice

James A Rickard[1,2], Holly Anderton[1,2†], Nima Etemadi[1,2†], Ueli Nachbur[1,2], Maurice Darding[3], Nieves Peltzer[3], Najoua Lalaoui[1,2], Kate E Lawlor[2,4], Hannah Vanyai[2,5], Cathrine Hall[1,2], Aleks Bankovacki[1,2], Lahiru Gangoda[6], Wendy Wei-Lynn Wong[7], Jason Corbin[2,8], Chunzi Huang[9], Edward S Mocarski[9], James M Murphy[1,2], Warren S Alexander[2,8], Anne K Voss[2,5], David L Vaux[1,2], William J Kaiser[9], Henning Walczak[3], John Silke[1,2]*

[1]Cell Signalling and Cell Death Division, Walter and Eliza Hall Institute of Medical Research, Melbourne, Australia; [2]Department of Medical Biology, University of Melbourne, Parkville, Australia; [3]Centre for Cell Death, Cancer, and Inflammation, University College London, London, United Kingdom; [4]Inflammation Division, Walter and Eliza Hall Institute of Medical Research, Melbourne, Australia; [5]Development and Cancer Division, Walter and Eliza Hall Institute of Medical Research, Melbourne, Australia; [6]Department of Biochemistry, La Trobe University, Bundoora, Australia; [7]Department of Immunology, Institute of Experimental Immunology, University of Zurich, Zurich, Switzerland; [8]Cancer and Haematology Division, Walter and Eliza Hall Institute of Medical Research, Melbourne, Australia; [9]Department of Microbiology and Immunology, Emory Vaccine Center, Emory University School of Medicine, Atlanta, United States

*For correspondence: j.silke@latrobe.edu.au

†These authors contributed equally to this work

**Abstract** SHARPIN regulates immune signaling and contributes to full transcriptional activity and prevention of cell death in response to TNF in vitro. The inactivating mouse *Sharpin cpdm* mutation causes TNF-dependent multi-organ inflammation, characterized by dermatitis, liver inflammation, splenomegaly, and loss of Peyer's patches. TNF-dependent cell death has been proposed to cause the inflammatory phenotype and consistent with this we show *Tnfr1*, but not *Tnfr2*, deficiency suppresses the phenotype (and it does so more efficiently than *Il1r1* loss). TNFR1-induced apoptosis can proceed through caspase-8 and BID, but reduction in or loss of these players generally did not suppress inflammation, although *Casp8* heterozygosity significantly delayed dermatitis. *Ripk3* or *Mlkl* deficiency partially ameliorated the multi-organ phenotype, and combined *Ripk3* deletion and *Casp8* heterozygosity almost completely suppressed it, even restoring Peyer's patches. Unexpectedly, *Sharpin*, *Ripk3* and *Casp8* triple deficiency caused perinatal lethality. These results provide unexpected insights into the developmental importance of SHARPIN.

## Introduction

Chronic proliferative dermatitis mutation (*cpdm*) mice are deficient in SHARPIN (*Sharpin^cpdm/cpdm*: henceforth referred to as *Shpn^m/m*; protein: SHARPIN) and develop dermatitis, multi-organ pathology and an immunological phenotype including disrupted lymphoid architecture, splenomegaly, liver inflammation and a loss of Peyer's patches in the gut (*HogenEsch et al., 1993*, *1999*; *Seymour et al., 2007*). SHARPIN is required for normal tumour necrosis factor (TNF) receptor 1 (TNFR1)-mediated gene induction and prevention of TNF-mediated death of various cells, including epidermal keratinocytes, in vitro (*Gerlach et al., 2011*; *Ikeda et al., 2011*; *Tokunaga et al., 2011*). The dermatitis is characterized

**eLife digest** In response to an injury or infection, areas of the body can become inflamed as the immune system attempts to repair the damage and/or destroy any microbes or toxins that have entered the body. At the level of individual cells inflammation can involve cells being programmed to die in one of two ways: apoptosis and necroptosis.

Apoptosis is a highly controlled process during which the contents of the cell are safely destroyed in order to prevent damage to surrounding cells. Necroptosis, on the other hand, is not controlled: the cell bursts and releases its contents into the surroundings.

Inflammation is activated by a protein called TNFR1, which is controlled by a complex that includes a protein called SHARPIN. Mice that lack the SHARPIN protein develop inflammation on the skin and internal organs, even in the absence of injury or infection. However, it is not clear how SHARPIN controls TNFR1 to prevent inflammation. Rickard et al. and, independently Kumari et al. have now studied this process in detail.

Rickard et al. cross bred mice that lack SHARPIN with mice lacking other proteins involved in inflammation and cell death. The experiments show that apoptosis is the main form of cell death in skin inflammation, but necroptosis has a bigger role in the inflammation of internal organs.

Mice that lack both the apoptotic and necroptotic cell-death pathways can develop relatively normally, but they die shortly after birth if they also lack SHARPIN. Experiments on these mice could help us to understand how SHARPIN works.

by epidermal cell death marked by cleaved caspase-3-, -8- and -9-positive cells (*Ikeda et al., 2011*; *Liang and Sundberg, 2011*; *Potter et al., 2014*). Since the dermatitis and inflammatory phenotype were shown to be TNF dependent, and because the only TNF signaling output that was aberrantly increased in the absence of SHARPIN was cell death, we previously proposed TNF/TNFR1-mediated cell death to be causative of the *cpdm* phenotype (*Gerlach et al., 2011*). The role of neither TNFR1 nor cell death has been confirmed in vivo, however.

TNFR1 signaling typically involves the intracellular recruitment of TNFR1-associated death domain protein (TRADD), TNF receptor-associated factor 2 (TRAF2), cellular inhibitor of apoptosis (cIAPs), and receptor interacting protein kinase 1 (RIPK1) (*Silke, 2011*). The heterotrimeric linear ubiquitin chain assembly complex (LUBAC) of SHARPIN (also known as SIPL), HOIL-1 (RBCK1/RNF54) and HOIL-1L-interacting protein (HOIP; RNF31) (*Gerlach et al., 2011*; *Ikeda et al., 2011*; *Tokunaga et al., 2011*) is also recruited to the TNFR1 signaling complex. Here, it assembles a linear ubiquitin scaffold needed for full recruitment of the NF-κB essential modulator (NEMO)/NF-κB kinase subunit gamma (IKKγ)-containing IKK complex, which activates pro-survival NF-κB signaling. TNFR1-induced c-Jun N-terminal protein kinase (JNK) and p38 signaling is also regulated by LUBAC. SHARPIN deficiency blunts the TNFR1 pro-survival transcriptional signal and sensitizes cells to TNF-induced cell death. The E3 ligase activity of HOIP catalyzes the addition of linear ubiquitin to target proteins, and SHARPIN and HOIL-1 are key regulators of the stability and activity of HOIP (*Gerlach et al., 2011*). In addition to TNFR1, LUBAC has also been shown to regulate the transcriptional response from the interleukin-1 receptor (IL-1R), CD40, lymphotoxin beta receptor (LTβR), toll-like-receptor 4 (TLR4), and nucleotide-binding oligomerization domain-containing protein 2 (NOD2) receptor signaling complexes (*Schmukle and Walczak, 2012*). Deletion of *Il1rap*, needed for IL-1 signaling, has been reported to almost completely prevent the development of *cpdm* dermatitis (*Liang et al., 2010*). This suggests that IL-1R signaling is a significant driver of disease, but the effect of *Il1rap* deficiency on the rest of the *Shpn^{m/m}* phenotype was not reported.

*Cpdm* mice have prominent eosinophil infiltration into the skin; however, deletion of *Il5*, which dramatically reduces the number of cutaneous and circulating eosinophils, fails to ameliorate disease (*Renninger et al., 2010*). *Shpn^{m/m}Rag1^{−/−}* mice lacking functional lymphocytes develop dermatitis, indicating that T and B cell cells are not required for the skin phenotype (*Potter et al., 2014*). Furthermore, hematopoietic cell transfer with bone marrow and spleen cells from *cpdm* mice to syngeneic wild-type C57BL/Ka mice failed to transfer disease in mice 2 months post reconstitution. Finally, *cpdm* skin transplanted onto nude mice retained the donor dermatitis phenotype 3 months post transplant, while syngeneic healthy skin transplanted onto *cpdm* mice did not acquire the disease over the same time

(*HogenEsch et al., 1993*; *Gijbels et al., 1995*). Together these studies indicate that a skin-intrinsic defect in *cpdm* mice drives the inflammatory disease, however they do not rule out a role for the hematopoietic system in amplifying it.

Impaired pro-survival TNFR1 signaling can induce both caspase-8-dependent apoptotic and RIPK3- and mixed lineage kinase domain-like protein (MLKL)-dependent necroptotic cell death via a cytosolic death platform (*Micheau and Tschopp, 2003*; *He et al., 2009*; *Sun et al., 2012*; *Zhao et al., 2012*; *Murphy et al., 2013*). Necroptosis involves the release of cellular contents including potential damage-associated molecular patterns (DAMPs) such as mitochondrial DNA, high mobility group box 1 protein (HMGB1), IL-33, and IL-1α (*Kaczmarek et al., 2013*). By contrast, apoptosis is considered to be immunologically silent, although this is clearly context dependent because excessive apoptosis resulting from conditional epidermal deletion of the caspase inhibitor cFLIP can cause severe skin inflammation (*Panayotova-Dimitrova et al., 2013*). Caspase-8 can cleave both RIPK1 and RIPK3 and is needed to keep the necroptotic pathway in check (*Vandenabeele et al., 2010*; *Kaiser et al., 2011*; *Oberst et al., 2011*). Regulation of necroptotic signaling is crucial for skin homeostasis because deletion of either caspase-8, the caspase-8 adaptor protein FADD (Fas-associated protein with death domain), or RIPK1, leads to RIPK3- and MLKL-dependent epidermal hyperplasia and inflammation (*Kovalenko et al., 2009*; *Lee et al., 2009*; *Bonnet et al., 2011*; *Kaiser et al., 2011*; *Oberst et al., 2011*; *Dannappel et al., 2014*; *Dillon et al., 2014*; *Rickard et al., 2014*).

Although the precise factors that determine whether TNFR1 mediates apoptosis or necroptosis are unclear, high levels of RIPK3, loss of cIAPs, and CYLD-mediated deubiquitylation of RIPK1 appear conducive to necroptosis (*Silke and Vaux, 2014*). In addition to a crucial role in necroptosis, RIPK3 may also regulate inflammasome-induced IL-1ß production in the absence of IAPs or caspase-8 (*Vince et al., 2012*; *Kang et al., 2013*). Thus the effects of loss of RIPK3 on an inflammatory phenotype may not be due to loss of necroptotic cell death but to a less well-defined role in IL-1ß production. This complicates interpretation of the role of RIPK3 in a disease, particularly when IL-1 is pathogenic such as in *cpdm* dermatitis. MLKL is downstream of RIPK3 in necroptosis and appears not to be required for cytokine production in the same situations as RIPK3 (*Wong et al., 2014)*. Thus *Mlkl*−/− mice may provide an opportunity to disentangle the relative contribution of necroptosis and deregulated cytokine production in disease.

Here we provide genetic evidence in support of our hypothesis that TNFR1-induced cell death is a driver of the inflammatory disease in *cpdm* mice (*Gerlach et al., 2011*). We show that the *cpdm* phenotype is TNFR1 and cell-death dependent. *Ripk3* or *Mlkl* deficiency largely prevented *cpdm* liver inflammation and ameliorated the splenic phenotype and leukocytosis. Remarkably, given the skin-inflammation phenotype of skin-specific *Casp8* knock-out mice (*Kovalenko et al., 2009*; *Lee et al., 2009*), *Casp8* heterozygosity potently suppressed the inflammatory skin phenotype while leaving the systemic inflammation unaffected. Strikingly, combined *Ripk3* deficiency and *Casp8* heterozygosity completely prevented the *cpdm* dermatitis in all but one of the mice analyzed at 42 to 45 weeks of age, prevented liver inflammation and grossly restored splenic architecture. Surprisingly, given the importance of LTßR (also known as TNFRSF3) signaling in the formation of Peyer's patches (*De Togni et al., 1994*; *Koni et al., 1997*) and the role of SHARPIN in this pathway (*Tokunaga et al., 2011*), apparently normal Peyer's patches were also present in *Shpn^m/m^Casp8^+/-^Ripk3^−/−^* mice.

## Results

### *Shpn^m/m^* dermatitis is mediated by TNFR1, IL-1R to a lesser extent and not TNFR2

The dermatitis in *Shpn^m/m^* mice has previously been shown to be driven by TNF (*Gerlach et al., 2011*) and IL-1 signaling (*Liang et al., 2010*). Because the environment may influence the onset of the disease we wished to test the importance of TNF and IL-1 signaling in a head-to-head manner. Furthermore the relative contribution of TNFR1 and TNFR2 in *cpdm* dermatitis has not been determined. We therefore generated *Shpn^m/m^* mice deficient in *Tnfr1*, *Tnfr2* or *Il1r1* (*Figure 1A*). All the knock-out mouse strains used in this study have been backcrossed at least ten times onto C57BL/6, or were generated on the C57BL/6 background (*Mlkl*−/− mice). However, the *cpdm* mutation arose on a C57BL/Ka background (*HogenEsch et al., 1993*). To control for background modifier effects, we backcrossed C57BL/Ka *Shpn^m/m^* mice once or twice onto the C57BL/6 background, equivalent to the strategy we used in generating all our *Shpn^m/m^* compound knock-out strains to generate

$Shpn^{m/m}$ C57BL/6 control mice. These control mice developed the *cpdm* phenotype indistinguishably from the C57BL/Ka $Shpn^{m/m}$ mice, typically presenting with a visible skin phenotype by 5 to 6 weeks and invariably requiring euthanasia due to the dermatitis before 14 weeks of age. *Tnfr2* deletion did not ameliorate the dermatitis but $Shpn^{m/m}Tnfr1^{-/-}$ mice showed no outward signs of disease even after 35 weeks (*Figure 1A–C*). *Il1r1* deletion significantly delayed the appearance of dermatitis, with markedly reduced epidermal hyperplasia in 13-week-old $Shpn^{m/m}Il1r1^{-/-}$ mice compared with 12-week-old $Shpn^{m/m}$ mice (*Figure 1B,C*). However, by 19–20 weeks, $Shpn^{m/m}Il1r1^{-/-}$ mice typically developed disease and required euthanasia. *Tnfr1* deletion reduced periportal liver inflammation and partially ameliorated the splenic phenotype, but did not restore Peyer's patches, whilst *Tnfr2* and *Il1r1* deletion did not prevent pathology in any of these organs (*Figure 1B,D*).

## Keratinocyte cell death and dermal macrophage infiltration are early events in $Shpn^{m/m}$ dermatitis

To gain insight into the pathogenesis of *cpdm* dermatitis we assessed cytokine levels in skin lysates from 6-week-old *cpdm* and control mice using a BioPlex kit (BioRad) (*Figure 2A*) to determine which cytokines were elevated early in the disease process. TNF levels were slightly elevated (log scale, *Figure 2A*) and, consistent with reports of eosinophilia in *cpdm* mice (*HogenEsch et al., 1993*; *Gijbels et al., 1996*), IL-5 (a key inducer of eosinophil maturation) was also elevated. The monocyte and macrophage chemoattractant protein MCP-1 was significantly elevated in *cpdm* skin, as was the IL-12 p40 subunit. There was also a trend for macrophage inflammatory protein 1α (MIP-1α) levels to be elevated. Consistent with this, there was an increase in F4/80⁺ cells in the $Shpn^{m/m}$ dermis, and this was evident in patches at 3 weeks, before significant epidermal hyperplasia was present (*Figure 2B*). Ikeda et al. reported cleaved caspase-3-positive keratinocytes in 10-week-old $Shpn^{m/m}$ mice (*Ikeda et al., 2011*), and we found cleaved caspase-3-positive cells were already present in the epidermis of 3-week-old $Shpn^{m/m}$ mice, indicating that apoptosis is an early event in the dermatitis and occurs before significant hyperplasia (*Figure 2C*). There were slightly more cleaved caspase-3-positive cells in $Shpn^{m/m}$ livers than controls (almost exclusively in the infiltrating cells), however they were infrequent. An increased number of cleaved caspase-3-positive cells were detected in $Shpn^{m/m}$ spleens, but, again, were much less appreciable than in the epidermis (*Figure 2C*).

Reconstitution of wild-type mice with $Shpn^{m/m}$ bone marrow and/or spleen cells fails to transfer the disease (*HogenEsch et al., 1993*), suggesting a skin-intrinsic defect underlies *cpdm* dermatitis. These mice, however, were only followed for 8 weeks post reconstitution and this may not have allowed sufficient time for the dermatitis to develop. To test this possibility, we reconstituted wild-type mice with $Shpn^{m/m}$ bone marrow cells and followed them for 12 months. Reconstitution efficiency was high but the mice did not demonstrate any skin, liver, or spleen phenotype during this time (*Figure 3*). Collectively these results suggest that the hematopoietic system in isolation cannot cause $Shpn^{m/m}$ dermatitis, and that macrophages may play a role in the amplification of the disease, particularly given they can be a prominent source of TNF.

## TNF induces cell death with caspase-3 and -8 cleavage in primary keratinocytes, mouse dermal fibroblasts and myeloid cells in vitro

*Tnf* deletion prevents *cpdm* dermatitis and TNF kills $Shpn^{m/m}$ keratinocytes in vitro, suggesting that TNF-induced cell death drives the $Shpn^{m/m}$ skin phenotype (*Gerlach et al., 2011*). Consistent with this, TNF-induced cell death in $Shpn^{m/m}$ keratinocytes that was partially blocked by Q-VD-OPh or Nec-1 treatment (*Figure 4A*). $Shpn^{m/m}$ mouse dermal fibroblasts (MDFs) were also sensitive to TNF-induced cell death, and this death was almost completely blocked by the RIPK1 kinase inhibitor Necrostatin-1 (Nec-1), but not by the pan-caspase inhibitor Q-VD-OPh (*Figure 4B*). To test whether TNF could induce caspase cleavage we treated primary $Shpn^{m/m}$ keratinocytes and MDFs with TNF for up to 4 hr (*Figure 4C,D*, *Figure 4—figure supplement 1*). TNF treatment led to caspase-3 and -8 cleavage at 4 hr in $Shpn^{m/m}$, but not wild-type, keratinocytes. In $Shpn^{m/m}$ MDFs there was also caspase-3 and -8 cleavage, and, surprisingly, this was reduced by Nec-1, but not Q-VD, treatment (*Figure 4D*). In $Shpn^{m/m}$ keratinocytes and MDFs, the increased caspase-8 activity coincided with a marked processing of cFLIP, but other outcomes of the signaling pathway were less obviously affected. Together with the early presence of cleaved caspase-3 staining and the elevation of TNF in $Shpn^{m/m}$ skin, these results indicate that TNF-induced apoptosis may play a pathogenic role in the skin phenotype.

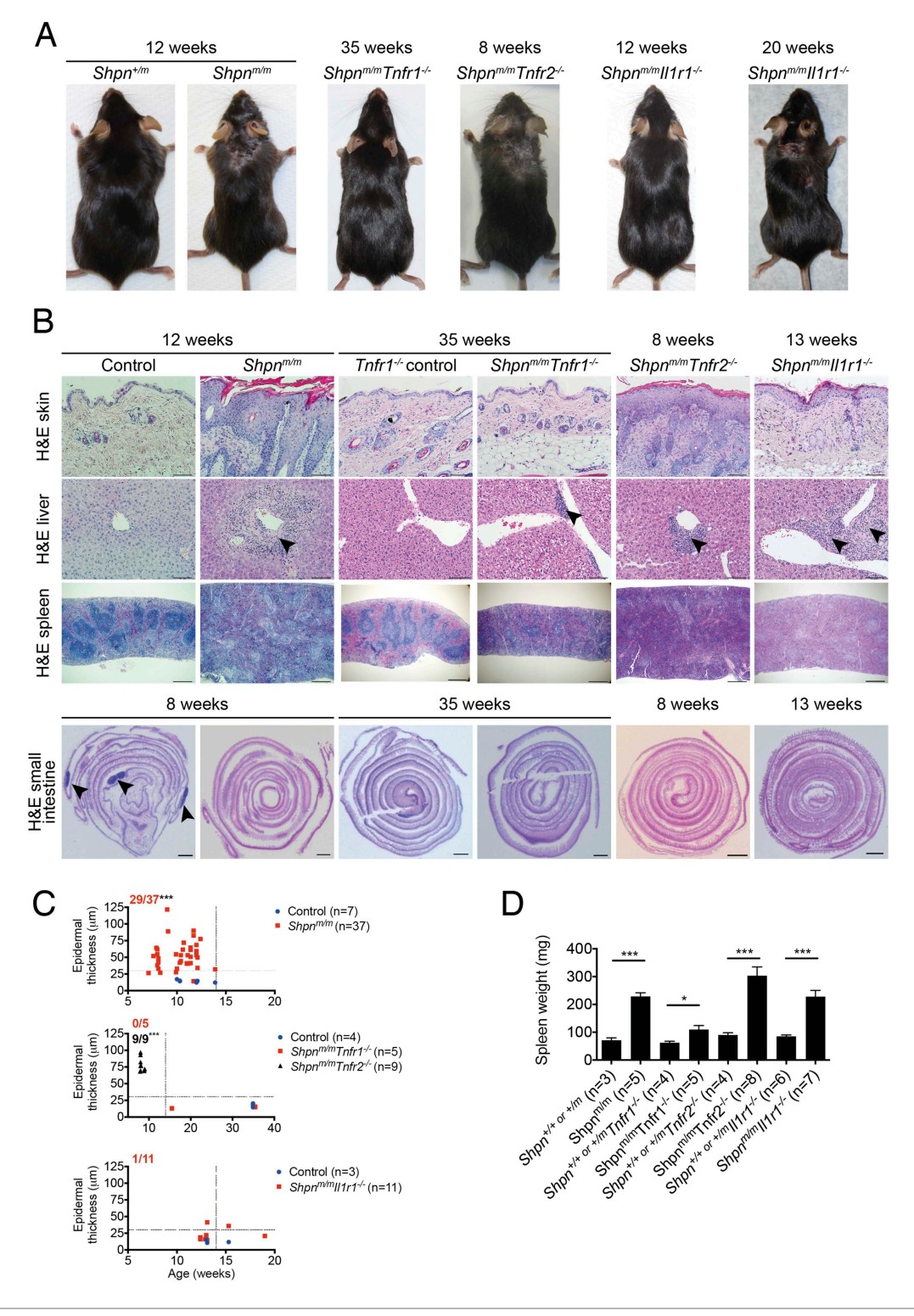

**Figure 1**. *Cpdm* dermatitis is mediated by TNFR1, IL-1R to a lesser extent, and not TNFR2. (**A**) Representative photos of mice of indicated genotypes and age. (**B**) Histological analysis of mice of genotype and age as indicated; representative of n ≥ 3 mice for each genotype or group. Black arrows in liver images point to areas of periportal inflammation. Black arrows in small intestine image point to Peyer's patches. *Shpn^{m/m}*: *Sharpin^{cpdm/cpdm}*. Control mice are *Shpn^{+/+ or +/m}*, *Tnfr1^{−/−}* control mice are *Shpn^{+/+ or +/m}Tnfr1^{−/−}*. Scale bars: Skin and liver 100 μm, spleen 500 μm,
*Figure 1. Continued on next page*

*Figure 1. Continued*

small intestine 1 mm. H&E: hematoxylin and eosin. (**C**) Epidermal thickness of mice of indicated age and genotypes. Each point represents the average of at least 14 measurements from multiple fields of view per mouse that were taken by a researcher who was blind to the genotype. Dotted lines are drawn at 30 µm and 14 weeks. Red numbers (and black for middle graph) correspond to proportion of $Shpn^{m/m}$ mice with epidermal thickness >30 µm at < 14 weeks of age (upper left quadrant). Control mice are $Shpn^{+/+ \text{ or } +/m}$, $Shpn^{+/+ \text{ or } +/m}Tnfr1^{-/-}$ and $Shpn^{+/+ \text{ or } +/m}Il1r1^{-/-}$ in upper, middle and lower graphs respectively. *** Significantly different to control group (Fisher's exact test), p ≤ 0.005. (**D**) Average spleen weights of mice of indicated genotypes. Spleen weights were taken from 12-week-old mice (except $Tnfr1^{-/-}$ mice that were 15 or 35 weeks old), or younger mice if they required euthanasia due to their dermatitis. Data are represented as mean + SEM, *p ≤ 0.05, ***p ≤ 0.005.

Given the appearance of cleaved caspase-3-positive cells in the $Shpn^{m/m}$ dermis (*Figure 2C*), we sought to determine whether some of these might be non-fibroblast cells such as immune cells. Since lymphoid cells are not required for *cpdm* dermatitis (*Potter et al., 2014*) and macrophage infiltration appears early, we tested whether myeloid cells were also sensitive to TNF-induced death. $Shpn^{m/m}$ neutrophils, monocytes and bone-marrow-derived macrophages (BMDMs) were all sensitive to killing by TNF, whereas the wild-type cells were not (*Figure 4E*). Neutrophil and monocyte cell death was more efficiently blocked by a combination of Nec-1 and Q-VD, whereas in macrophages Nec-1 was sufficient to block cell death. These data suggest that myeloid cell death may contribute to the inflammatory skin phenotype.

## The catalytic LUBAC component HOIP, like SHARPIN, is required to protect keratinocytes against TNF-induced death

We did not detect a defect in p38, JNK or NF-κB pro-survival signaling in SHARPIN-deficient keratinocytes and MDFs, as has been shown for other cell types (*Gerlach et al., 2011*; *Ikeda et al., 2011*; *Tokunaga et al., 2011*). Consistent with earlier reports (*Gerlach et al., 2011*; *Peltzer et al., 2014*), however, using an antibody that specifically recognises linear ubiquitin we detected substantially reduced linear ubiquitylation in the native TNFR1 signaling complex obtained from $Shpn^{m/m}$ vs wild-type mouse embryonic fibroblasts (MEFs) (*Figure 5A*). We also found that HaCaT keratinocytes (a human immortalized cell line) stably expressing catalytically inactive HOIP$^{C885S}$ were sensitive to TNF-induced cell death (*Figure 5B*), indicating a requirement for LUBAC and its linear-ubiquitin-chain-forming activity in preventing TNF-induced keratinocyte death.

## *Casp8* heterozygosity, but not *Bid* deficiency, delays onset of *cpdm* dermatitis

To investigate the importance of the caspase-8-mediated apoptotic pathway in vivo we generated $Shpn^{m/m}Casp8^{+/-}$ mice. We did not attempt to generate $Shpn^{m/m}Casp8^{-/-}$ mice because *Casp8* deletion results in embryonic lethality at around E10.5 (*Varfolomeev et al., 1998*). In contrast to $Shpn^{m/m}$ mice, 12-week-old $Shpn^{m/m}Casp8^{+/-}$ mice had almost no epidermal hyperplasia and largely normal keratin 6 and 14 expression (*Figure 6A,B*). By 15 weeks of age, however, significant hyperplasia was observed in one of these mice (*Figure 6B*). $Shpn^{m/m}Casp8^{+/-}$ mice retained other aspects of the phenotype: splenomegaly with disrupted splenic architecture, liver inflammation, and a lack of intestinal Peyer's patches (*Figure 6C*). Some $Shpn^{m/m}Casp8^{+/-}$ mice succumbed to a pulmonary infection with *Pasteurella* and required euthanasia while littermates were unaffected, suggesting that $Shpn^{m/m}Casp8^{+/-}$ mice are partially immunocompromised. Mice with obvious signs of infection were not used for the spleen, blood, or liver analyses but were included in the skin analysis. Occult infection, however, in $Shpn^{m/m}Casp8^{+/-}$ mice used for hematopoietic, liver, and spleen analyses cannot be completely excluded.

In certain cell types such as hepatocytes, BID can be cleaved by caspase-8 to generate truncated BID (tBID) that, in turn, mediates caspase-9- and mitochondria-dependent apoptosis (*Czabotar et al., 2014*). BID is a key regulator of UV-induced apoptosis in keratinocytes, indicating that this pathway is of importance in the skin (*Pradhan et al., 2008*). Because cleaved caspase-9 is found in the $Shpn^{m/m}$ epidermis (*Ikeda et al., 2011*) and $Shpn^{m/m}$ keratinocyte extracts contained caspase-9-substrate-cleaving activity (*Liang and Sundberg, 2011*), we hypothesized that TNF-induced BID-dependent apoptosis may be a driver of the *cpdm* phenotype and, hence, generated $Shpn^{m/m}Bid^{-/-}$ mice. We

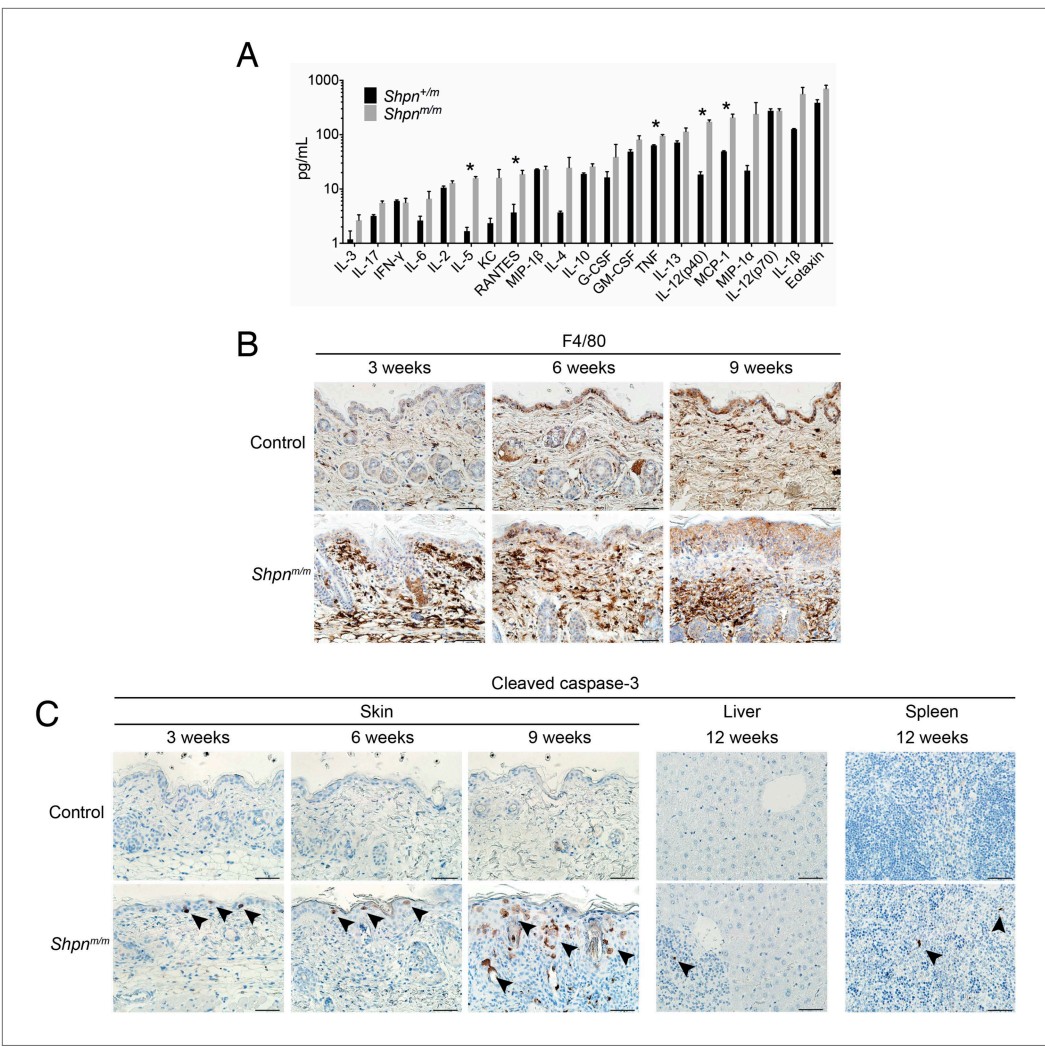

**Figure 2**. Keratinocyte cell death and dermal macrophage infiltration are early events in *cpdm* dermatitis.
(**A**) BioPlex cytokine analysis of skin lysates from mice of indicated genotypes. Data are represented as mean
+S.E.M. *p ≤ 0.05. (**B**) F4/80 staining (brown) of skin sections counterstained with hematoxylin (blue). Control:
*Shpn⁺/⁺ or ⁺/ᵐ*. (**C**) Cleaved caspase-3 staining (brown) of skin, liver and spleen sections counterstained with
hematoxylin (blue). Black arrows show examples of cleaved-caspase-3 positive cells. Control: *Shpn⁺/⁺ or ⁺/ᵐ*.
(**A–C**) Three mice were analyzed for each genotype or group. Scale bars: 50 μm.

observed no protection from any aspects of the *cpdm* phenotype (**Figure 6A–D**), however, indicating that caspase-8, but not BID, is an important driver of this disease.

### *Ripk3* deletion slightly delays *Shpnᵐ/ᵐ* dermatitis, and *Ripk3* and *Mlkl* deletion partially protects against the *cpdm* splenic phenotype and markedly attenuates liver inflammation

Caspase-8 heterozygosity significantly delayed epidermal hyperplasia, but other aspects of the *cpdm* phenotype (e.g. splenomegaly and liver inflammation) remained. We therefore sought to test the role of necroptosis in the inflammatory disease by generating *Shpnᵐ/ᵐRipk3⁻/⁻* mice. Whereas control mice invariably developed severe dermatitis by 12 weeks of age, roughly half of the 12-week-old *Shpnᵐ/ᵐRipk3⁻/⁻* mice had a less severe epidermal phenotype at this point (**Figure 7A,B**). When aged over 12 weeks, all *Shpnᵐ/ᵐRipk3⁻/⁻* animals went on to develop severe disease and were euthanized due to their skin phenotype before 18 weeks of age. The remainder of *Shpnᵐ/ᵐRipk3⁻/⁻* mice developed skin disease at the same rate as control mice. 12 week-old *Shpnᵐ/ᵐRipk3⁻/⁻* mice had no signs of liver inflammation and significantly less splenomegaly, although they still lacked Peyer's

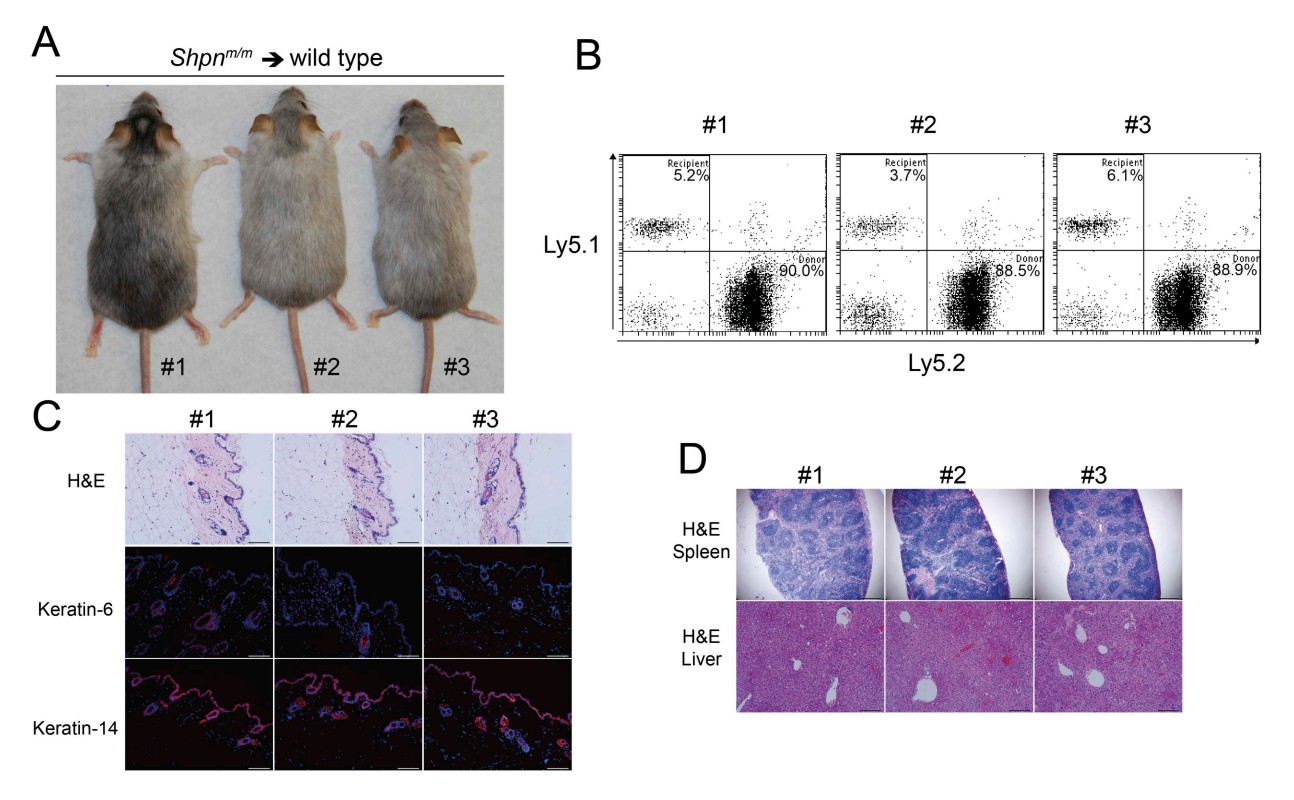

**Figure 3**. No dermatitis in wild-type mice reconstituted with Shpn*m/m* bone marrow after 12 months. (**A**) No dermatitis was observed in wild-type mice 12 months after reconstitution with *Shpn*m/m* bone marrow. (**B**) Flow cytometry analysis showing percentage contribution of Ly5.1 (recipient) vs Ly5.2 (donor) white blood cells in peripheral blood 12 months after reconstitution. (**C**) Histological and immunofluorescence analysis of dorsal skin from mice 12 months after reconstitution. (**D**) Histological analysis of the spleen and liver from mice 12 months after reconstitution. Scale bars: Skin 100 μm, liver 200 μm and spleen 500 μm. H&E: hematoxylin and eosin.

patches (*Figure 7C,D*). We also generated *Shpn*m/m*Mlkl*−/−* mice and found they had a similar epidermal phenotype to *Shpn*m/m* mice (*Figure 7B*). Like *Shpn*m/m*Ripk3*−/−* mice, however, 12-week-old *Shpn*m/m*Mlkl*−/−* animals had reduced splenomegaly and only one out of 12 mice showed signs of liver inflammation (*Figure 7C,D*). Collectively these results indicate that RIPK3 and MLKL are important drivers of the liver and splenic *cpdm* phenotype, and that RIPK3 contributes to the epidermal phenotype, importantly in a non-MLKL-dependent (hence, most likely, necroptosis-independent) manner.

### *Shpn*m/m*Casp8*-/-*Ripk3*−/−* mice are prone to perinatal lethality

Since deletion neither of one allele of *Casp8* nor of two alleles of *Ripk3* was able to fully rescue all the multi-organ pathology in *cpdm* mice, we sought to generate *Shpn*m/m*Casp8*−/−*Ripk3*−/−* mice, taking advantage of the fact that *Ripk3* deletion prevents *Casp8*−/−*-mediated embryonic lethality (*Kaiser et al., 2011*; *Oberst et al., 2011*). We generated these mice independently at two separate facilities. At one facility, most of these triple-deficient mice died perinatally, typically in a window between E17 and 1 to 2 days after birth, and no *Shpn*m/m*Casp8*-/-*Ripk3*−/−* animals were obtained at weaning (*Figure 8A,B*; numbers in A refer to mice analyzed at first facility only). Most *Shpn*m/m*Casp8*−/−*Ripk3*−/−* embryos obtained by Caesarean section at E19 appeared edematous or were dead. Some were able to breathe; however, these died within the first 2 days after birth. Hematopoietic analysis at E19 did not reveal any consistent differences (*Figure 8C*). At a separate facility, two viable *Shpn*m/m*Casp8*−/−*Ripk3*−/−* mice were obtained from a limited number of matings. By roughly 3 months of age these mice appeared runted and were euthanized. Histologically these mice had no epidermal phenotype, but disrupted splenic architecture was apparent (*Figure 8D*).

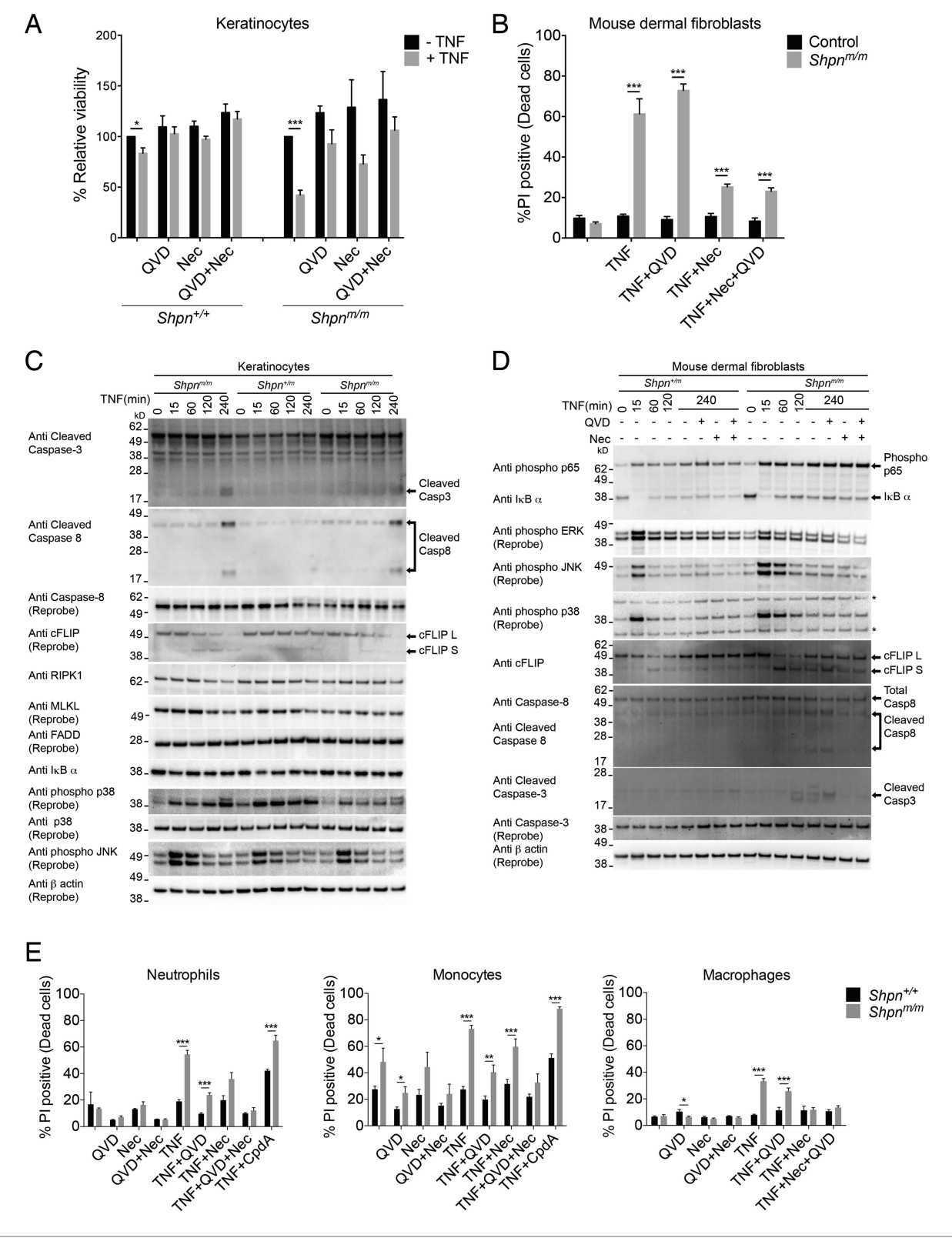

**Figure 4**. TNF induces death in multiple primary cell types, and is marked by caspase-3 and -8 cleavage in primary keratinocytes and mouse dermal fibroblasts in vitro. (**A** and **B**) Primary keratinocytes (**A**) and mouse dermal fibroblasts (MDFs) (**B**) were treated with 100 ng/ml human Fc-TNF, 50 µM Nec-1 or 10 µM Q-VD-OPh for 24 hr as indicated, then viability was assessed by propidium iodide (PI) uptake and flow cytometry. Cells were generated from
*Figure 4. Continued on next page*

*Figure 4. Continued*

three different mice for each group. Control: *Shpn*[+/+ or +/m]. (**C** and **D**) Western blot analysis of primary keratinocytes (**C**) and MDFs (**D**) treated as indicated (concentrations as in **A** and **B**) then lysed and lysates separated on SDS/PAGE and western blotted with indicated antibodies. *Shpn*[+/m]: n = 1; *Shpn*[m/m]: n = 2 (n = 1 for MDFs) mice analyzed shown above. * Indicates non-specific band. Data from additional mice is shown in *Figure 4—figure supplement 1*. (**E**) Neutrophils, monocytes and bone-marrow-derived macrophages were treated with 100 ng/ml human Fc-TNF, 50 μM Nec-1, 20 μM Q-VD-OPh (10 μM for macrophages) or 500 nM Compound A (CpdA) for 20 hr (24 hr for macrophages) as indicated. Viability was assessed by PI uptake and flow cytometry. The Smac mimetic CpdA sensitizes cells to TNF-induced cell death and serves as a control. Cells were generated from three different mice for each group except for macrophages, where six to eight mice were analyzed. (**A**, **B**, **E**) Data are represented as mean + SEM, *$p \leq 0.05$, **$p \leq 0.01$, ***$p \leq 0.005$.

The following figure supplement is available for figure 4:

**Figure supplement 1**. TNF induces caspase-3 and -8 cleavage in primary keratinocytes and mouse dermal fibroblasts in vitro.

### *Casp8* heterozygosity and *Ripk3* deletion markedly delays *Shpn*[m/m] phenotype

While the reason for the lethality of the triple-deficient animals is unknown, we were readily able to obtain viable *Shpn*[m/m]*Casp8*[+/−]*Ripk3*[−/−] mice. These mice were indistinguishable from control mice at 12 weeks of age and had no skin, liver, or spleen pathology (*Figure 9A–E*). Remarkably, Peyer's patches were present (*Figure 9C*, *Figure 9—figure supplement 1*). One 67-week-old and three 45-week-old *Shpn*[m/m]*Casp8*[+/−]*Ripk3*[−/−] mice had no signs of any skin phenotype, however another mouse developed dermatitis at 42 weeks (*Figure 9A,B,D*; note that epidermal thickness was not quantified for the 42-week-old mouse).

### *Shpn*[m/m] leukocytosis is largely mediated by *Ripk3* and *Mlkl*

To investigate the effect of the various genetic crosses on *Shpn*[m/m] leukocytosis, we analyzed white blood cell levels in peripheral blood using an ADVIA hematological analyzer (*Figure 10*). *Tnfr1* deletion was more effective at reducing leukocyte numbers than *Tnfr2* or *Il1r1* deletion, although all of these compound knock-out mice had leukocyte subsets that were elevated compared to controls. *Caspase-8* heterozygosity or *Bid* deletion did not prevent the leukocytosis, whereas *Ripk3* and *Mlkl* deletion markedly reduced it, suggesting the hematopoietic phenotype is driven predominantly by necroptosis. *Shpn*[m/m]*Ripk3*[−/−] and *Shpn*[m/m]*Mlkl*[−/−] mice still had elevated neutrophils, however *Shpn*[m/m]*Casp8*[+/−]*Ripk3*[−/−] mice had no white blood cell elevation.

## Discussion

LUBAC, composed of three proteins, HOIL-1, HOIP and SHARPIN, has recently emerged as a regulator of a diverse set of signaling complexes (*Zak et al., 2011*; *Schmukle and Walczak, 2012*; *Tokunaga, 2013*). Lack of HOIP causes mid-gestational embryonic lethality (embryonic day 10.5), and this is prevented by simultaneous loss of *Tnfr1* (*Peltzer et al., 2014*). Together, these data demonstrate that LUBAC is required for full-strength IL-1ß and TNF signaling and deficiency in these signaling pathways would be expected to impair an inflammatory response. The consequence, however, of *Sharpin* deficiency in mice is multi-organ inflammation (*HogenEsch et al., 1993*). Likewise, loss-of-expression and loss-of-function mutations in HOIL-1 result in a fatal, human, inherited disorder characterized by chronic inflammation; consistent with loss of the major defensive inflammatory signaling pathways, affected patients suffer from invasive bacterial infections (*Boisson et al., 2012*). In vitro experiments suggest that HOIL-1 is an essential component of LUBAC, which is present in many cell types. With this in mind it is surprising that HOIL-1-deficient mice were reported to be normal (*Tokunaga et al., 2009*). If, however, we set this observation to one side, it seems that in humans and mice loss of LUBAC can precipitate an inflammatory disease. This could be because the loss of the homeostatic component of TNFR1, IL-1R1, and TLR signaling reduces the ability of an organism to resist infection, which, even in the weakened signaling environment of LUBAC deficiency, drives inflammatory signaling. Another possibility is that full-strength signaling from TNF, IL-1ß, or TLR ligands is required to upregulate essential negative feedback regulators such as IκBα and A20. Signaling, therefore, although reduced, is constitutively active. This explanation seems less likely because IκBα and A20 transcripts were still upregulated in response to TNF or Pam$_3$CSK$_4$ in HOIL-1/HOIP-deficient and *cpdm* cells (*Haas et al., 2009*; *Tokunaga et al., 2009*; *Gerlach et al., 2011*; *Zak et al., 2011*).

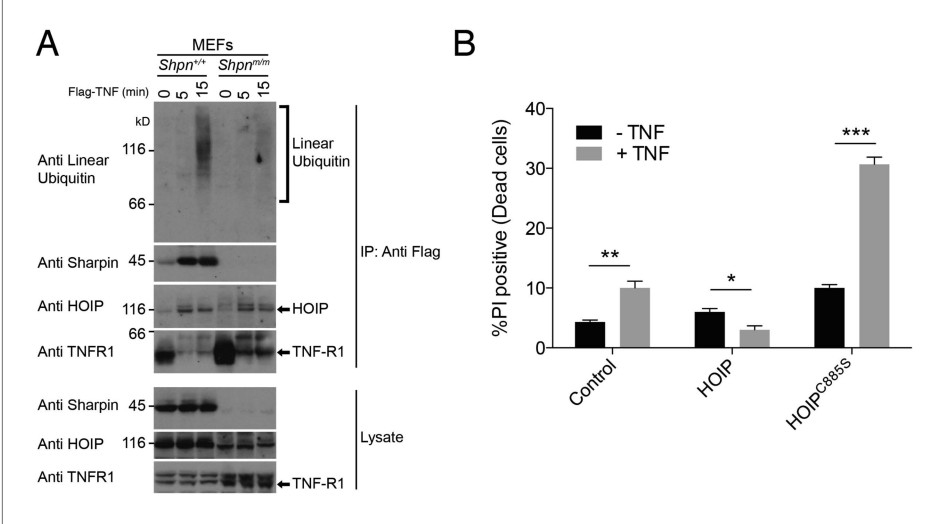

**Figure 5**. *Sharpin* is required for normal linear ubiquitylation of the TNF-R1 signaling complex, and HOIP protects keratinocytes from TNF-induced cell death. (**A**) Anti Flag Immuno-Precipitation (IP) of the TNF receptor signaling complex in immortalised mouse embryonic fibroblasts (MEFs) treated with Flag-TNF for the times indicated. (**B**) HaCaT human keratinocytes stably expressing HOIP, the catalytically inactive HOIP[C885S], or an empty vector (control) were treated with 100 ng/ml TNF for 24 hr. Viability was assessed by propidium iodide (PI) uptake and flow cytometry. Data are presented as mean + SEM, n = 3, *p ≤ 0.05, **p ≤ 0.01, ***p ≤ 0.005.

The fact that the phenotype from the *cpdm* mice could not be transferred by the hematopoietic compartment, and that the skin phenotype is maintained following skin transplantation onto nude mice, indicates that these animals suffer from an intrinsic skin defect (*HogenEsch et al., 1993*; *Gijbels et al., 1995*). This is not unprecedented; loss of Notch signaling in the skin causes a dermatitis disease in mice that is sufficient to drive systemic inflammation with similar features to the *cpdm* phenotype (*Dumortier et al., 2010*). The earlier hematopoietic reconstitution experiments, however, may not have allowed sufficient time for the inflammatory phenotype to develop. The work described here excludes this caveat because reconstituted mice did not develop signs of the *cpdm* phenotype even a year after reconstitution; thus, the inflammatory skin phenotype in the *Shpn*[m/m] mutant mice is the result of an intrinsic skin defect. It is, however, noteworthy that loss of *Mlkl* did not affect the *Shpn*[m/m] skin phenotype but significantly reduced or prevented splenomegaly and liver inflammation. Conversely, reduction in caspase-8 markedly ameliorated the skin phenotype but did not prevent splenomegaly or liver inflammation. This shows that the skin and systemic phenotype are separable, although the exact mechanism is unclear.

The fact that *cpdm* keratinocytes are sensitized to TNF-induced apoptosis and necroptosis led us to hypothesize that TNF/TNFR1-dependent cell death was causative for the skin phenotype (*Gerlach et al., 2011*). Because of the systemic inflammation we suspected that necroptotic cell death and the release of DAMPs was the driver. Consistent with this hypothesis, we show here that *Tnfr1* deficiency (as with *Tnf* deficiency) suppresses the *cpdm* phenotype, whereas *Tnfr2* deficiency had no effect on the phenotype. Loss of IL-1 signaling has been shown to suppress the *cpdm* phenotype (*Liang et al., 2010*), but because the environment of the mice likely plays a large part in the onset and severity of the inflammatory disease it was not possible to determine whether loss of IL-1 signaling is as potent as *Tnf* deficiency at preventing the phenotype. We therefore generated *Shpn*[m/m]*Il1r1*[−/−] mice, which succumbed to the inflammatory disease much later than *Shpn*[m/m] mice, but were far more inflammation prone than any of the *Shpn*[m/m]*Tnf*[−/+], *Shpn*[m/m]*Tnf*[−/−], or *Shpn*[m/m]*Tnfr1*[−/−] mice. These data are again consistent with our original hypothesis, suggesting that TNF/TNFR1 signaling is the main driver of inflammation and that IL-1 contributes to exacerbating the disease.

*Casp8* deficiency results in early embryonic lethality in mice (*Varfolomeev et al., 1998*), and loss of *Casp8* or *Fadd* in the skin leads to keratinocyte hyperplasia and inflammatory skin disease (*Kovalenko et al., 2009*; *Bonnet et al., 2011*). Therefore, we considered it unlikely that loss of caspase-8 would

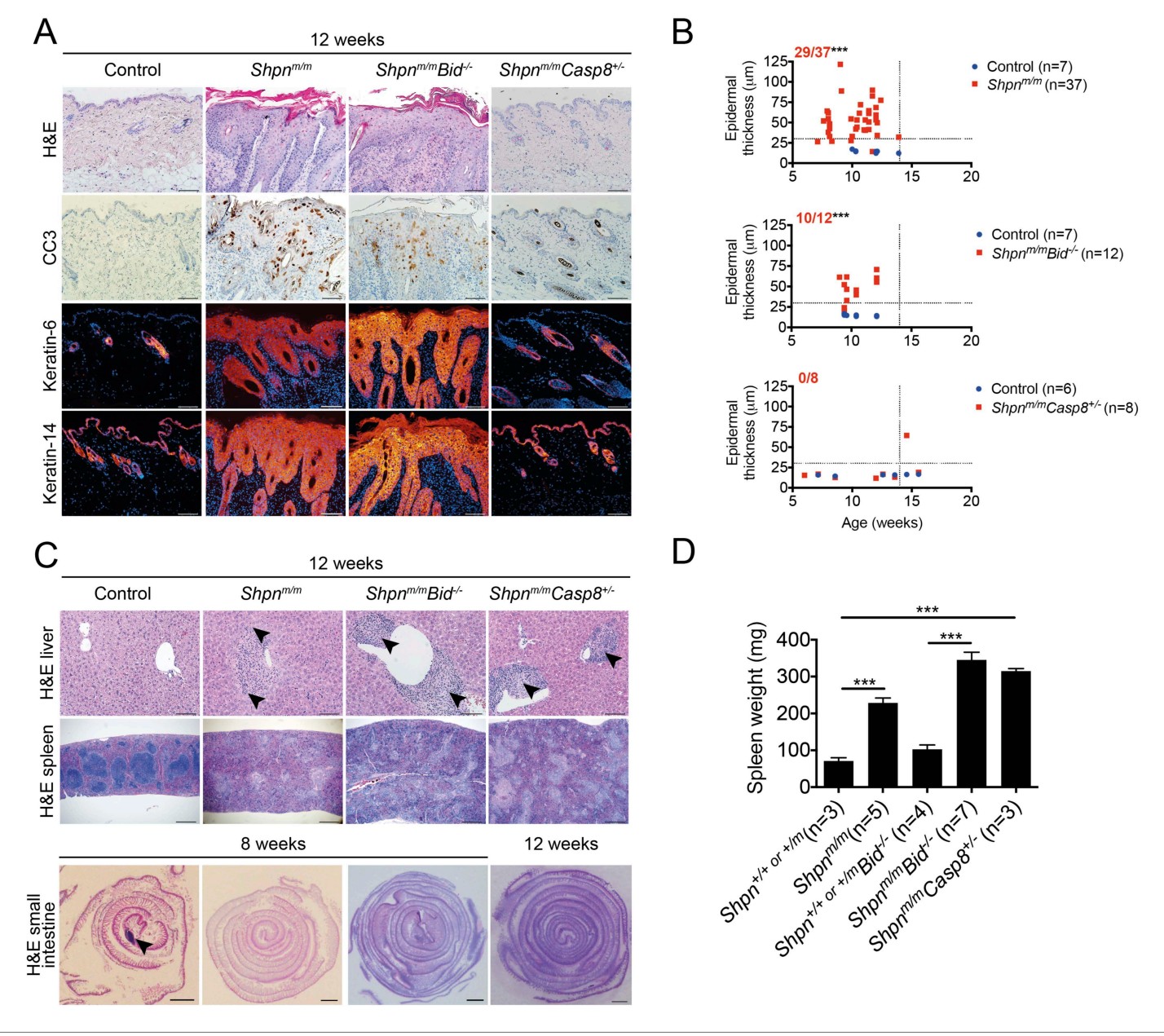

**Figure 6**. Protection from *cpdm* dermatitis with *Casp8* heterozygosity but not *Bid* deletion. (**A**) Histological and immunofluorescence skin analysis. (**B**) Epidermal thickness of mice of indicated age and genotypes determined as in *Figure 1B*, by an investigator blinded to genotype. Top panel is a repeat of data in 1B for reference purposes only. Dotted lines are drawn at 30 µm and 14 weeks. Red numbers correspond to proportion of *Shpn^m/m* mice with epidermal thickness >30 µm at < 14 weeks of age (upper left quadrant). Control mice are *Shpn^+/+ or +/m*, *Shpn^+/+ or +/m Bid^-/-*, and *Shpn^+/+ or +/m Casp8^+/-* in upper, middle and lower graphs, respectively. *** Significantly different to control group (Fisher's exact test), p ≤ 0.005. (**C**) Histological analysis of spleen, liver, and small intestine. Black arrows in liver images point to areas of periportal inflammation. Black arrow in small intestine image points to Peyer's patch. (**D**) Average spleen weights of mice of indicated genotypes. Spleen weights were taken from 12–14-week-old mice, or younger mice if they required euthanasia due to their dermatitis. Data are represented as mean + SEM, ***p ≤ 0.005. (**A** and **C**) Control mice are *Shpn^+/+ or +/m*, n ≥ 3 mice analyzed each genotype or group. Scale bars: skin and liver 100 µm, spleen 500 µm, small intestine 1 mm. H&E: hematoxylin and eosin.

diminish the chronic proliferative dermatitis of the *Shpn^m/m* mutant mice. To our surprise, however, loss of a single allele of *Casp8* was strikingly effective at reducing the skin phenotype and appearance of cleaved caspase-3 in the skin. It has been proposed that the intrinsic mitochondrial apoptosis pathway mediated by caspase-9 and caspase-3 plays a role in the keratinocyte hyperplasia observed

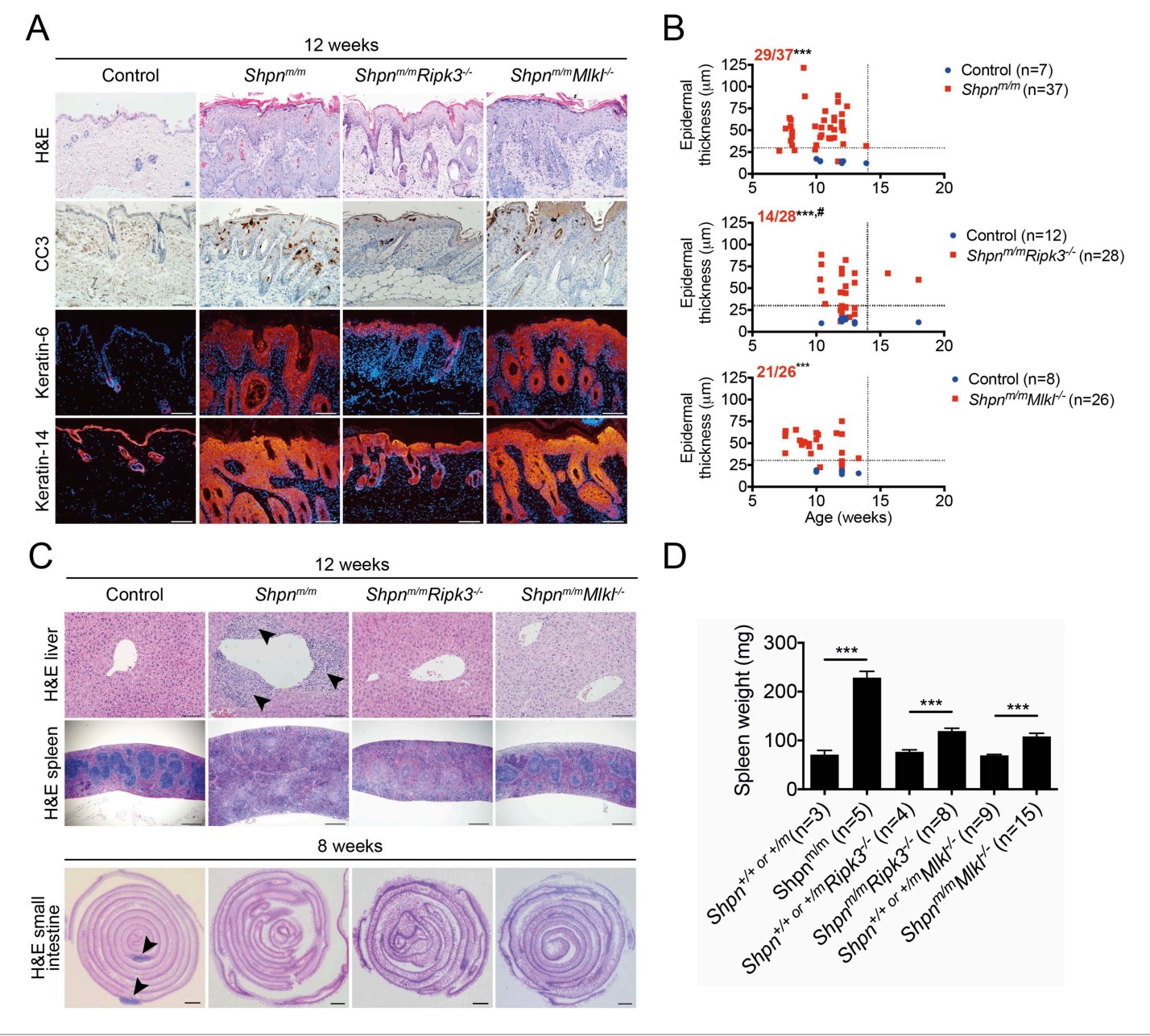

**Figure 7**. *Ripk3* deficiency slightly delays *cpdm* dermatitis onset, and *Ripk3* and *Mlkl* deficiency partially protects against the *cpdm* splenic phenotype and markedly attenuates liver inflammation. (**A**) Histological and immunofluorescence skin analysis. (**B**) Epidermal thickness of mice of indicated age and genotypes determined as in ***Figure 1B***, by an investigator blinded to genotype. Top panel is a repeat of data in 1B for reference purposes only. Dotted lines are drawn at 30 μm and 14 weeks. Red numbers correspond to proportion of *Shpn^{m/m}* mice with epidermal thickness >30 μm at < 14 weeks of age (upper left quadrant). Control mice are *Shpn^{+/+ or +/m}*, *Shpn^{+/+ or +/m}Ripk3^{−/−}*, and *Shpn^{+/+ or +/m}Mlkl^{−/−}* in upper, middle and lower graphs, respectively. *** Significantly different to control group (Fisher's exact test), p ≤ 0.005, # significantly different to *Shpn^{m/m}* mice (Fisher's exact test), p ≤ 0.05. (**C**) Histological analysis of spleen, liver, and small intestine. Black arrows in liver image points to areas of periportal inflammation. Black arrows in small intestine image points to Peyer's patches. (**D**) Average spleen weights of mice of indicated genotypes. Spleen weights were taken from 12-week-old mice, or younger mice if they required euthanasia due to their dermatitis. Data are represented as mean + SEM, ***p ≤ 0.005. (**A** and **C**) Control mice are *Shpn^{+/+ or +/m}*, n ≥ 3 mice analyzed each genotype or group. Scale bars: skin and liver 100 μm, spleen 500 μm, small intestine 1 mm. H&E: hematoxylin and eosin.

in *Shpn^{m/m}* mutant mice based on the presence of disrupted mitochondria in *cpdm* skin and in vitro experiments (***Liang and Sundberg, 2011***). Given the prominent role that caspase-8 plays in the dermatitis, it would be expected that if the intrinsic apoptosis pathway is engaged it should be

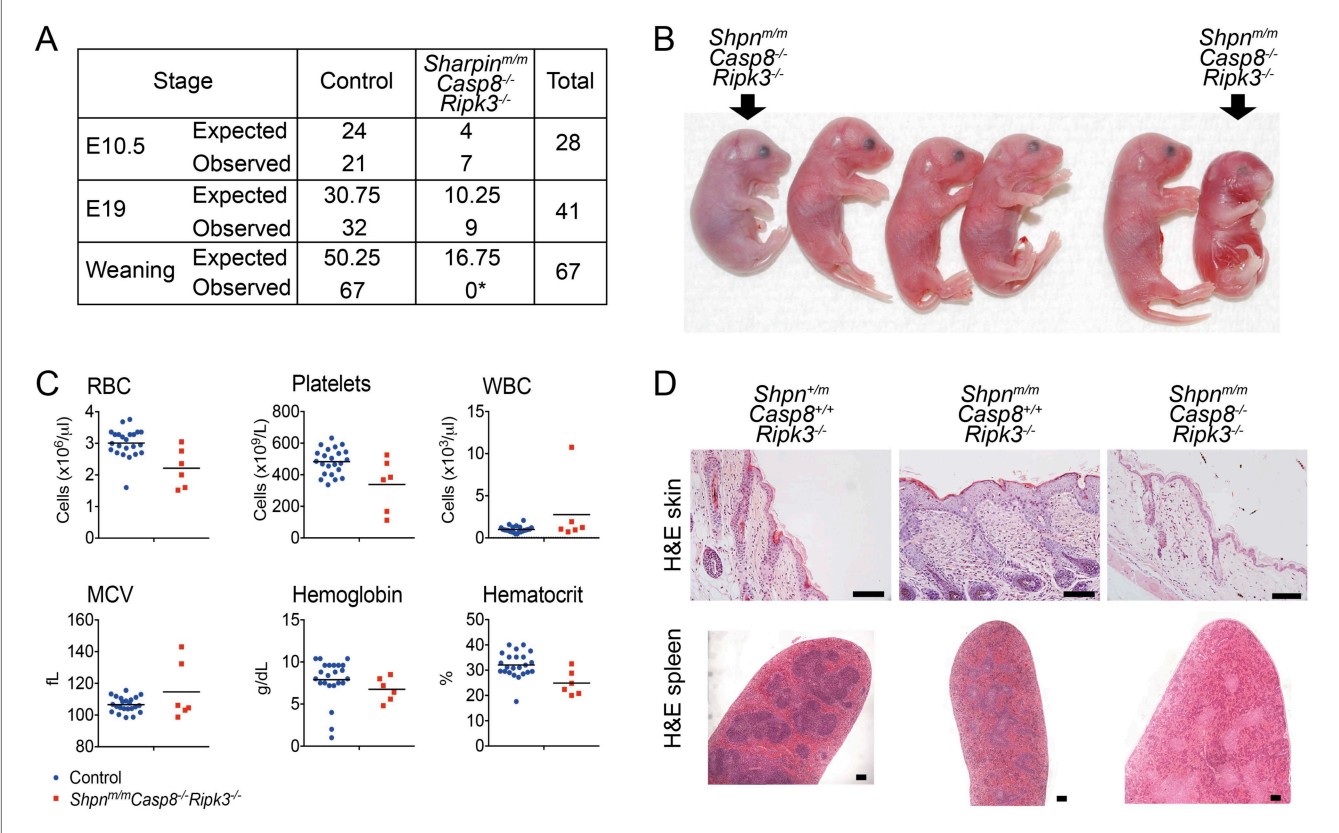

**Figure 8**. Shpn^m/m^Casp8^−/−^Ripk3^−/−^ mice are prone to perinatal lethality. (**A**) Table of segregation of expected and observed genotypes from *Shpn^+/m^Casp8^-/^Ripk3^−/−^* or *Shpn^m/m^Casp8^+/-^Ripk3^−/−^* intercrosses at various developmental stages. E10.5 controls: *Shpn^+/+ or +/m^Casp8^-/^Ripk3^−/−^*; E19 controls: *Shpn^m/m^Casp8^+/+ or +/−^Ripk3^−/−^*; weaning controls: *Shpn^+/+ or +/m^Casp8^-/−^Ripk3^−/−^* or *Shpn^m/m^Casp8^+/+ or +/−^Ripk3^−/−^*. *Significantly different to expected value, Fisher's exact test p < 0.0005. (**B**) Photos of E19 embryos obtained by Caesarian section. The *Shpn^m/m^Casp8^−/−^Ripk3^−/−^* mouse on the right was recovered dead, all others were alive. Other embryos are control mice: *Shpn^m/m^Casp8^+/+ or +/−^Ripk3^−/−^*. (**C**) ADVIA blood analysis from E19 embryos. RBC: red blood cells; WBC: white blood cells; MCV: mean cell volume. Horizontal lines depict data means. Control mice: *Shpn^m/m^Casp8^+/+ or +/−^Ripk3^−/−^*. (**D**) Histological analysis of tissue from 12-week-old mice of indicated genotypes. Mice were from a separate facility to those in **A**–**C** and are not included in the table in **A**. Two mice were analyzed for each genotype. Scale bars: skin 100 µm, spleen 50 µm.

downstream of caspase-8 and require cleavage of the BH3 protein, BID (*Czabotar et al., 2014*). Cells that require BID cleavage by caspase-8 in order to undergo apoptosis are known as type II cells (*Barnhart et al., 2003*), but it is unclear whether keratinocytes are type I or type II (*Pradhan et al., 2008*; *Geserick et al., 2014*). Genetic deletion of *Bid* did not suppress the dermatitis or any other aspect of the *cpdm* phenotype, therefore we conclude that the intrinsic mitochondrial pathway is unlikely to play a significant role in the disease.

Because necroptotic cell death can be inflammatory, by provoking the release of DAMPs, we expected that deficiency in RIPK3 or MLKL, essential effectors of the necroptotic cell death pathway, would reduce the severity of the inflammation in *Shpn^m/m^* mice. Whilst there was a modest delay in the appearance of the dermatitis in *Shpn^m/m^Ripk3^−/−^* mice, all animals went on to develop severe skin disease. In contrast to RIPK3 deficiency, deletion of *Mlkl* did not even delay appearance of the dermatitis. Together, this suggests that RIPK3 may exacerbate the skin phenotype independently of necroptosis, possibly via a direct role in cytokine production. Combined with the markedly ameliorated dermatitis seen in *Shpn^m/m^Casp8^+/−^* mice, this indicates that apoptosis is the main driver of *Shpn^m/m^* dermatitis. Although apoptosis is generally regarded as being immunologically inert, when in excess it can result in severe inflammation. For example, conditional deletion of *cFLIP* in adult keratinocytes caused severe inflammation in the epidermis (*Panayotova-Dimitrova et al., 2013*). Like the *cpdm* phenotype, this inflammation was TNF dependent. An excess of apoptotic cells may cause disease by

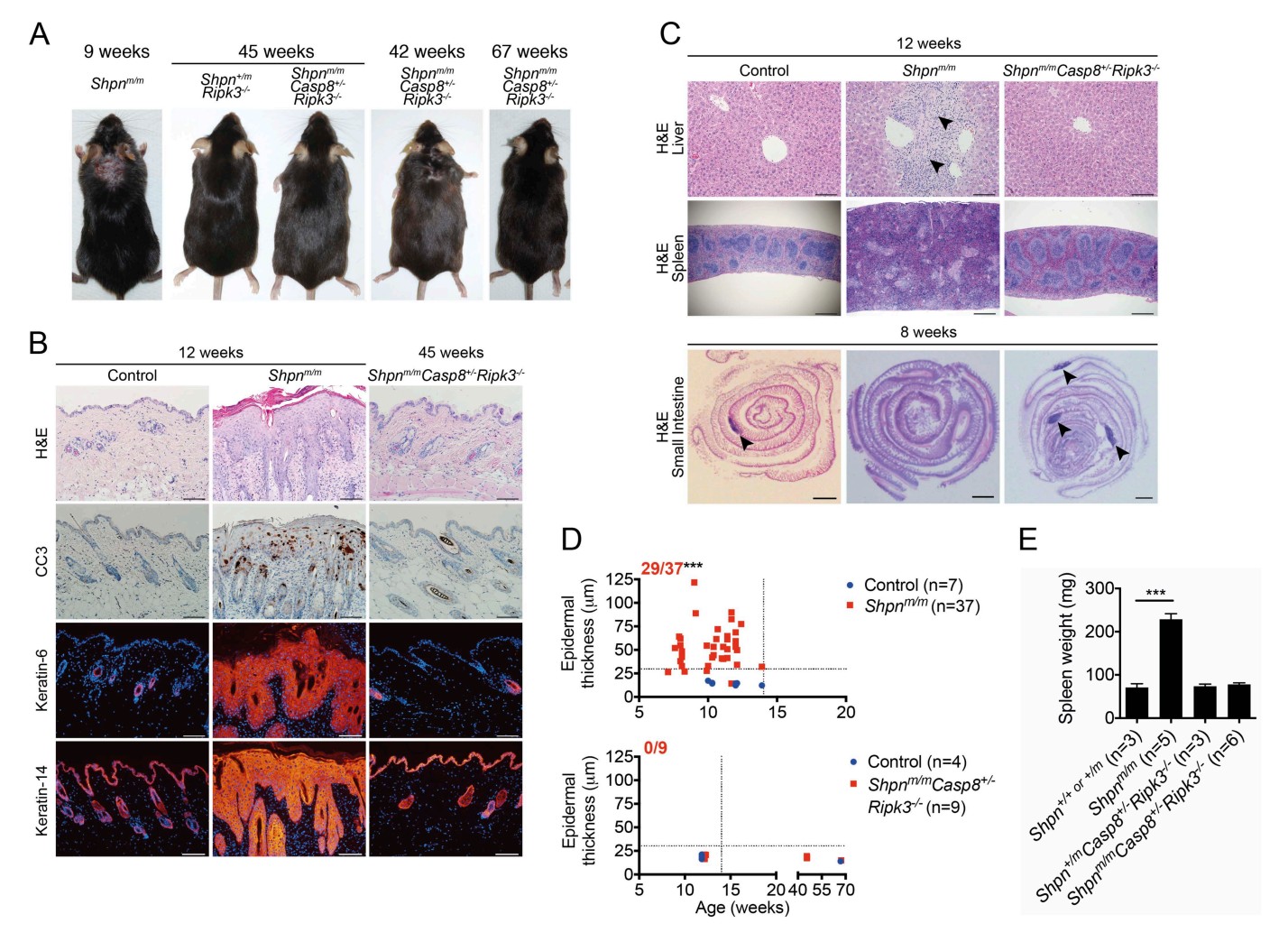

**Figure 9**. *Ripk3* deletion and Casp8 heterozygosity markedly delays emergence of *cpdm* dermatitis and liver inflammation and restores Peyer's patches. (**A**) Representative photos of mice of indicated genotypes and age. Three *Shpn^{m/m}Casp8^{+/-}Ripk3^{-/-}* mice were analyzed at 45 weeks and one at 67 weeks with no detectable dermatitis, however another mouse developed dermatitis by 42 weeks of age. (**B**) Histological and immunofluorescence skin analysis. (**C**) Histological analysis of spleen, liver, and small intestine. Black arrows in liver image point to areas of periportal inflammation. Black arrows in small intestine images point to Peyer's patches. (**D**) Epidermal thickness of mice of indicated age and genotypes measured as described in *Figure 1B*. Top panel is a repeat of data in 1B for reference purposes only. Dotted lines are drawn at 30 μm and 14 weeks. Red numbers correspond to proportion of *Shpn^{m/m}* mice with epidermal thickness >30 μm at < 14 weeks of age (upper left quadrant). Control mice are *Shpn^{+/+ or +/m}* and *Shpn^{+/+ or +/m}Casp8^{+/-}Ripk3^{-/-}* in upper and lower graphs, respectively. *** Significantly different to control group (Fisher's exact test), p ≤ 0.005. (**E**) Average spleen weights of mice of indicated genotypes. Spleen weights were taken from 12-week-old mice, or younger mice if they required euthanasia due to their dermatitis. Data are represented as mean + SEM, ***p ≤ 0.005. (**B** and **C**) Control mice are *Shpn^{+/+ or +/m}*, n ≥ 3 mice analysed each genotype or group. Scale bars: skin and liver 100 μm, spleen 500 μm, small intestine 1 mm. H&E: hematoxylin and eosin.

The following figure supplement is available for figure 9:

**Figure supplement 1**. Restoration of Peyer's patches in 8-week-old *Shpn^{m/m}Casp8^{+/-}Ripk3^{-/-}* mice.

overwhelming phagocytosis and clearance of apoptotic bodies, leading to secondary necrosis and DAMP release. It has also been proposed that in certain contexts, such as viral infections, apoptosis may be inflammatory (*Cullen et al., 2013*). In other *Shpn^{m/m}* organs such as the liver, spleen, and hematopoietic system, *Ripk3* and *Mlkl* deletion ameliorated or protected against disease, indicating that *Sharpin* deficiency triggers apoptosis in some tissues but necroptosis in others. Because it has been shown that RIPK3 or MLKL can mediate cytokine production induced by the absence of caspase-8

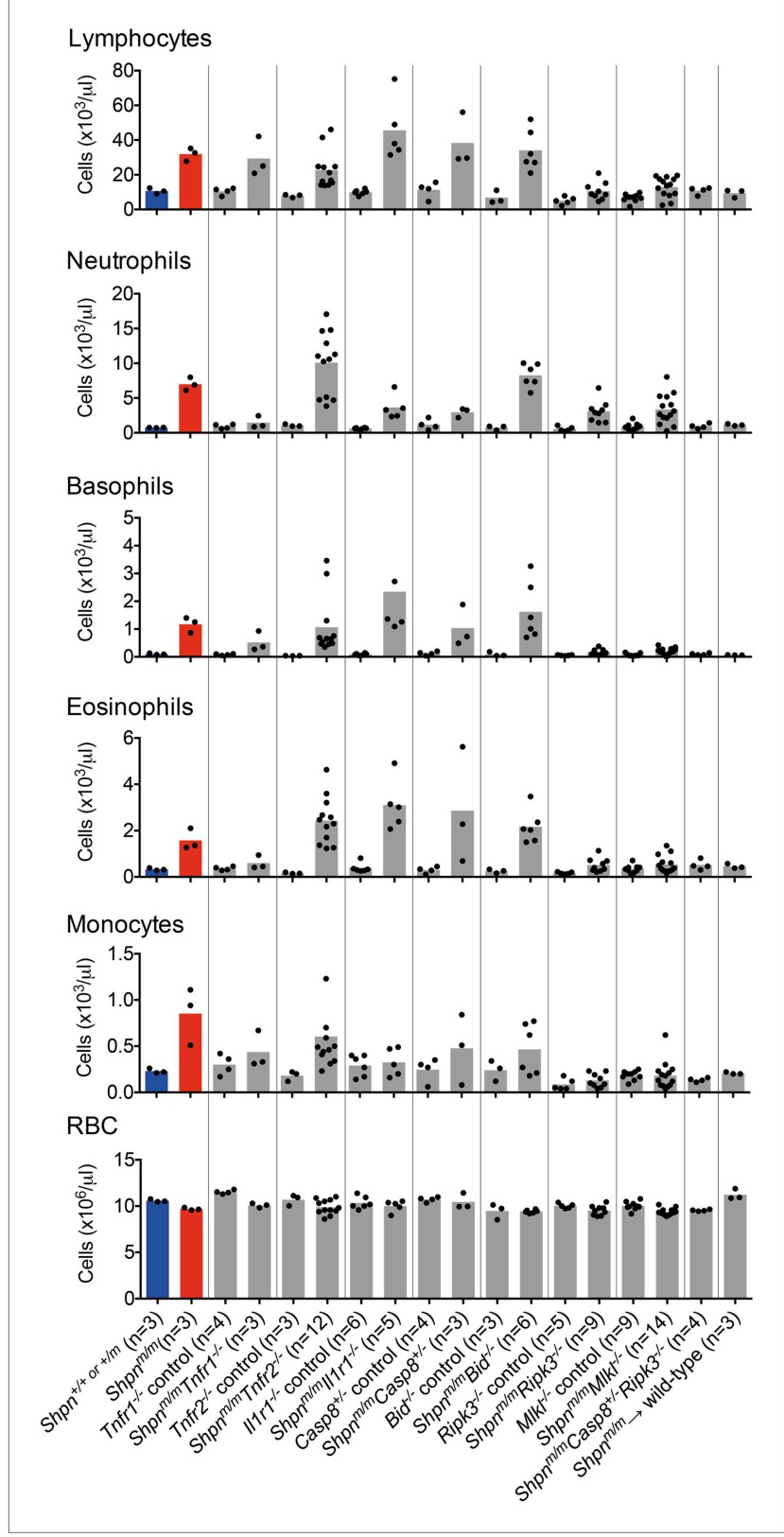

**Figure 10**. Peripheral blood counts from various crosses. Peripheral blood was collected from 11–14-week-old mice (reconstituted mice: *Shpn*^*m/m* → wild type were 12 months old and *Tnfr1*^−/− mice were 35 weeks old), or younger
*Figure 10. Continued on next page*

*Figure 10. Continued*

if the mice were euthanized due to severe dermatitis. $Shpn^{+/+\ or\ +/m}$ and $Shpn^{m/m}$ mice are highlighted in blue and red for reference purposes. Blood was analyzed using an ADVIA 2120 hematological analyzer. RBC: red blood cells. Horizontal lines depict data means. Control mice are $Shpn^{+/+\ or\ +/m}$, for example $Tnfr1^{-/-}$ control mice are $Shpn^{+/+\ or\ +/m}Tnfr1^{-/-}$.

(*Kang et al., 2013*), and potentially other, as yet undescribed, pathways, we cannot completely exclude the possibility that *Ripk3* or *Mlkl* deletion affords protection by blunting cytokine production. Absence of SHARPIN, however, leads to increased, not decreased, caspase-8 activity, so it is not obvious whether these observations apply in this case. Furthermore, whereas it is clear that RIPK3 plays a role in promoting cytokine production in response to a number of stimuli (*Vince et al., 2012*), we have so far only observed a defect in necroptosis and not in inflammatory cytokine production in $Mlkl^{-/-}$ cells (*Murphy et al., 2013*; *Allam et al., 2014*; *Rickard et al., 2014*).

Whilst $Shpn^{m/m}Casp8^{+/-}$, $Shpn^{m/m}Ripk3^{-/-}$, and $Shpn^{m/m}Mlkl^{-/-}$ mice all developed significant disease in either the liver, spleen, hematopoietic compartment, or skin, $Shpn^{m/m}Casp8^{+/-}Ripk3^{-/-}$ mice were almost completely protected from disease for approximately 10–11 months, and in one case 15 months. One of these mice developed the skin phenotype at 10 months of age, indicating that the single allele of *Casp8* is enough to eventually cause dermatitis. This is supported by the development of epidermal phenotype in 14–15-week-old $Shpn^{m/m}Casp8^{+/-}$ mice. SHARPIN has been shown to regulate LTß/LTßR signaling, and LTßR is required for the development of Peyer's patches (*De Togni et al., 1994*), therefore we had assumed that the absence of Peyer's patches was due to defective LTßR signaling. A recent study, however, demonstrated that rudimentary Peyer's patches formed in $Shpn^{m/m}$ embryos, but that these regressed post-natally (*Seymour et al., 2013*). Our data show that whilst neither *Casp8* heterozygosity nor *Ripk3* deletion in isolation restored intestinal Peyer's patches, the combination was able to do so. This suggests that the post-natal regression of these secondary lymphoid organs in $Shpn^{m/m}$ mice is due to deregulated cell death.

In a wider sense, our results are particularly important when interpreting the recent finding that crossing *cpdm* mice with $Ripk1^{K45A/K45A}$ mice (which lack RIPK1 kinase activity) completely suppresses the *cpdm* phenotype (*Berger et al., 2014*). RIPK1 has a promiscuous role in immune signaling, regulating pro-survival, necroptotic, and apoptotic pathways. However, while the RIPK1 kinase domain is well known for its ability to activate the necroptotic pathway, it is not believed to be required to cause apoptosis (*Vandenabeele et al., 2010*). Yet we show here that the $Shpn^{m/m}$ skin phenotype is associated with the appearance of processed caspase-3 in the skin, and both the epidermal hyperplasia and appearance of processed caspase-3 is markedly reduced by loss of a single allele of *Casp8*. Consistent with the hypothesis that $Shpn^{m/m}$ keratinocyte hyperplasia is due to apoptosis, it is not prevented by loss of the key necroptotic effector MLKL. Furthermore, purified $Shpn^{m/m}$ keratinocytes and dermal fibroblasts rapidly activate caspase-8 and caspase-3 in response to TNF. Taken together with the work of Berger et al. our work inescapably suggests that in the context of the $Shpn^{m/m}$ epidermis the dominant role of the RIPK1 kinase domain is to activate apoptosis. Unexpectedly, but supporting this conclusion, we showed that Nec-1 (the RIPK1 kinase inhibitor) blocked TNF-induced caspase-8 and caspase-3 activation in $Shpn^{m/m}$ dermal fibroblasts. This conclusion is particularly confronting because we and others have shown that loss of RIPK1 in the skin results in a RIPK3/MLKL-dependent hyperplasia that is presumably dependent on necroptosis (*Dannappel et al., 2014*; *Rickard et al., 2014*). Thus RIPK1 is able to both activate and inhibit either apoptosis or necroptosis in a highly context-dependent manner.

*Ripk3* deletion prevents the embryonic lethality seen with either *Fadd* or *Casp8* deletion and the mice survive to adulthood (*Kaiser et al., 2011*; *Oberst et al., 2011*; *Dillon et al., 2012*). Given this, the perinatal lethality we observed in $Shpn^{m/m}Casp8^{-/-}Ripk3^{-/-}$ mice was completely unexpected. At E19 some of these mice were recovered alive by Caesarean section and established regular breathing, some failed to establish normal breathing, some appeared edematous and were not recovered alive, whilst others were in the process of being resorbed from earlier embryonic lethality. To complicate the picture, two viable $Shpn^{m/m}Casp8^{-/-}Ripk3^{-/-}$ mice were obtained at a separate facility. At 3 months of age these mice had no epidermal phenotype but were runted. One potential explanation for the different penetrance of the phenotype may be a different genetic background because the mice described in *Figure 8A* were exon 3-deleted caspase-8 mice (*Beisner et al., 2005*), whereas the mice

in *Figure 8D* were exon 3- and 4-deleted caspase-8 (*Salmena et al., 2003*). Exon 3- and 4-deleted caspase-8 mice in yet a third facility (not shown), however, did not survive past weaning, indicating that environmental differences undoubtedly also contribute to the variable penetrance. Future efforts aimed at understanding the lethality of *Shpn^{m/m}Casp8^{−/−}Ripk3^{−/−}* mice should yield important insights into not only the biology of SHARPIN and linear ubiquitin, but also that of caspase-8 and RIPK3.

In summary, we provide strong evidence that *Sharpin* deficiency sensitizes keratinocytes to TNF/TNFR1-induced, caspase-8-mediated apoptosis, and that this defect appears to drive the *cpdm* dermatitis. *Ripk3* deletion provided only a modest and variable delay in the presentation of dermatitis. Unlike *Casp8* heterozygosity, *Ripk3* and *Mlkl* deletion ameliorated many aspects of the systemic *cpdm* phenotype. This indicates a tissue-specific role for *Sharpin* in regulating cell death pathways. Only combined *Casp8* heterozygosity and *Ripk3* deficiency was able to almost completely prevent all aspects of the *cpdm* systemic phenotype that we evaluated, including the early loss of Peyer's patches. Whilst inflammation is a known sequelae to necroptotic DAMP release, these findings provide further evidence that excessive apoptosis can also cause inflammatory disease. Furthermore, these results indicate that the suppression of *cpdm* dermatitis seen by crossing to mice lacking RIPK1 kinase activity (*Berger et al., 2014*) may, surprisingly, be due to RIPK1's kinase activity being upstream of caspase-8 in mediating TNF-induced apoptosis.

## Materials and methods

### Mice

Mice were maintained at the Walter and Eliza Hall Institute of Medical Research (WEHI) and University College London (UCL). C57BL/Ka *Sharpin^{cpdm/cpdm}* mice were obtained from Jax (Bar Harbor, ME), and then either backcrossed one to two times onto C57BL/6 or crossed with C57BL/6 *Ripk3^{−/−}*, *Mlkl^{−/−}*, *Casp8^{+/−}*, *Casp8^{+/-}Ripk3^{−/−}*, *Bid^{−/−}*, *Il1r1^{−/−}*, *Tnfr1^{−/−}*, or *Tnfr2^{−/−}* mice. For timed matings, mice were analyzed by Caesarean section. For E19 timed matings, pregnant females were injected at E17 and E18 with progesterone.

### Cell culture and western blotting

Primary keratinocytes and MDFs were isolated and cultured as described previously (*Gerlach et al., 2011*; *Etemadi et al., 2013*). Cell lysates were prepared using DISC buffer (1% NP-40, 10% glycerol, 150 mM NaCl, 20 mM Tris pH 7.5, 2 mM EDTA, cOmplete protease inhibitor cocktail (Roche; Penzberg, Germany), 2 mM sodium orthovanadate, 10 mM sodium fluoride, β-glycerophosphate, $N_2O_2PO_7$). Cell lysates were loaded in NuPAGE Bis-Tris gels (Life Technologies/Thermo Fisher Scientific; Waltham, MA) and transferred on to Immobilon-P PVDF membranes (Millipore; Billerica, MA) or Hybond-C Extra (GE Healthcare; Little Chalfont, UK). Membranes were blocked and antibodies diluted in 5% skim milk powder or Bovine Serum Albumin (BSA) in 0.1% PBS or TBS-Tween20. Antibodies used for western blot: cleaved caspase-3 (9661) and −8 (8592), phospho-JNK1/2 (4668P), phospho-p38 (4511), p38 (9212), caspase-8 (4927), JNK1/2 (9252), IκBα (CN: 9242), and phospho-p65 (3033) from Cell Signaling Technology (Danvers, MA), β-actin (A-1978; Sigma Aldrich; St. Louis, MO), RIPK1 (610458; BD Biosciences; Franklin Lakes, NJ), cFLIP (AG-20B-0005; Adipogen; Liestal, Switzerland), FADD (generated in-house; gift from Lorraine O'Reilly) and MLKL (generated in house; *Murphy et al., 2013*). Signals were detected by chemoluminescence (Millipore) after incubation with secondary antibodies conjugated to horseradish peroxidase.

For isolation of neutrophils and monocytes, red blood cells were lysed and bone marrow cells were stained with flurochrome-conjugated anti-mouse Ig antibodies (CD11b [Mac-1] and Ly6G [1A8]) and sorted using a FACS ARIA instrument (BD Biosciences). Neutrophils (CD11b⁺ Ly6G⁺) and monocytes (CD11b⁺Ly6G⁻) were cultured in 5% FCS RPMI at $1 \times 10^5$ and $0.5 \times 10^5$ cells per well, respectively, in a 96-well u-bottom tissue culture plate. BMDMs were isolated and cultured as described previously (*Wong et al., 2014*).

### Death assays

Keratinocytes, MDFs, neutrophils, monocytes, and BMDMs were stimulated with TNF (100 ng/ml), Nec-1 (50 µM), QVD-Oph (10 µM, 20 µM for neutrophils and monocytes) and CpdA (911, 500 nM). After 24 hr (20 hr for neutrophils and monocytes) all cells except keratinocytes were stained with propidium iodide (PI) and cell death analyzed on a FACScalibur instrument (BD Biosciences). For the keratinocyte MTS viability assay phenazine methosulfate (PMS; 0.92 mg/ml in PBS; Sigma-Aldrich) and 3-(4,5-dimethylthiazol-2-yl)-5-(3-carboxymethoxyphenyl)-2-(4-sulfophenyl)-2H-tetrazolium (MTS;

2 mg/ml in PBS; Promega; Fitchburg, WI) were combined in a 1:20 ratio. The mixture was added to cell culture media in a 1:5 ratio and incubated for 1–4 hr at 37°C in a humidified 5% $CO_2$ incubator. Media was transferred to a flat-bottom 96-well plate and absorbance was measured at 490 nm. Viability was calculated relative to the untreated sample.

For HaCaT experiments, cells stably expressing HOIP, HOIP$^{C885S}$ (catalytically inactive), or an empty vector were seeded and incubated the next day with 100 ng/ml histidine-tagged TNF for 24 hr. Supernatant and adherent cells were harvested and resuspended in PBS containing 5 mg/ml PI. PI-positive cells were measured by flow cytometry (BD Accuri; BD Biosciences).

### Histology and immunofluorescence

Paraffin-embedded tissue was fixed in 10% neutral buffered formalin then processed and stained with hematoxylin and eosin (H&E) according to standard practices. Immunohistochemistry and immunofluorescence analysis was performed as described previously (*Rickard et al., 2014*).

### Reconstitution experiments

Ly5.1 mice were irradiated with 2 × 550 rads spaced 3 hr apart. Following red blood cell lysis, ~ 5 × 10$^6$ BM cells from *Shpn*$^{m/m}$ (Ly5.2) mice were intravenously injected. Mice were maintained on 2 mg/ml neomycin in drinking water for 3 weeks post irradiation. Reconstitution efficiency was assessed 6 weeks and 12 months post reconstitution by staining for Ly5.1 and Ly5.2 in peripheral blood obtained from retro-orbital bleeding.

### Cytokine BioPlex assay

Cytokines were analyzed using a BioPlex Pro mouse cytokine 23-plex kit (Bio-Rad; Hercules, CA), or for analysis of TNF levels a mouse TNF ELISA kit (eBioscience; San Diego, CA) was used. Skin lysates were prepared by homogenizing skin in ice cold protein lysis buffer (20 mM Tris pH 7.5, 150 mM NaCl, 2 mM EDTA, 1% Triton-X100, 10% glycerol) using a Tissue Lyser II (QIAGEN; Hilden, Germany) for 12 cycles of 30 s at 30 Hz. A BCA kit (Thermo Fisher Scientific) was used to normalise protein levels. Values below the reference range were assigned the value of the lowest standard.

### ADVIA blood analysis

Peripheral blood was collected into EDTA tubes (Sarstedt; Nümbrecht, Germany) and analyzed using an ADVIA 2120 hematological analyzer.

### TNF-RSC immunoprecipitation

For TNF-RSC isolation, immortalised MEFs were stimulated with 3xFlag-2xStrep-TNF at 0.5 µg/ml for the indicated times, or left untreated. Cells were subsequently solubilized in lysis buffer (30 mM Tris–HCl (pH 7.4), 150 mM NaCl, 2 mM EDTA, 2 mM KCl, 10% Glycerol, 1% Triton X100, EDTA-free proteinase inhibitor cocktail (Roche) and 1x phosphatase inhibitor cocktail 2 (Sigma Aldrich)) at 4°C for 30 min. The lysates were cleared by centrifugation, and 3xFlag-2xStrep-TNF was added to the untreated samples. Next, lysates were subjected to anti-Flag immunoprecipitation using M2 beads (Sigma Aldrich) for 16 hr. The beads were washed three times with lysis buffer, proteins were eluted in 50 µl of LDS buffer (Life Technologies/Thermo Fisher Scientific) containing 50 mM DTT. Samples were analyzed by western blotting. Antibodies used were: HOIP (custom-made, Thermo Fisher Scientific), SHARPIN (14626-1-AP; ProteinTech; Chicago, IL), TNFR1 (ab19139; Abcam; Cambridge, UK), and linear ubiquitin (Genentech; South San Francisco, CA).

### Statistics

Pearson chi-square and Fisher's exact test were used to assess frequencies of observed vs expected genotypes during development, at birth, and at weaning. Fisher's exact test was used for epidermal thickness statistical calculations. Student's *t* test was used to calculate statistical significance shown for all other data.

## Acknowledgements

We thank staff in The Walter and Eliza Hall Institute of Medical Research (WEHI) and UCL Bioservices facilities, the WEHI Histology department and FACS lab, Vishva Dixit for *Ripk3*$^{-/-}$ mice, Vishva Dixit and Domagoj Vucic for the linear-ubiquitin-specific antibody, Heinrich Korner for *Tnf*$^{-/-}$, *Tnfr1*$^{-/-}$, and *Tnfr2* $^{-/-}$ mice, Stephen Hedrick and Razqallah Hakem for *Casp8*$^{fl/fl}$, Philippe Bouillet for *Bid*$^{-/-}$, and Dr M Labow for *Il1r1*$^{-/-}$ mice. We thank Julia Zinngrebe for technical assistance and George Varigos for discussions and support. This work was supported by the Thomas William and Violet Coles

Trust Fund, NHMRC grants (1016647, 461221, 1025594, 1046984, 1046010, 1051210, 1057905), APA scholarships (JAR, HV), ARC Fellowship (JMM) and NHMRC fellowships to AKV, JS and WSA (575512, 541901, 1058190, 1058344), NIH grant to WJK (DP1 OD012198) and ESM (NIH (US PHS grant R01 GM112547)), a Wellcome Trust Senior Investigator Award (096831/Z/11/Z) and an ERC Advanced Grant (294880) to HW with additional support from the Australian Cancer Research Fund, Victorian State Government Operational Infrastructure Support and NHMRC IRIISS grant (361646).

## Additional information

### Funding

| Funder | Grant reference number | Author |
|---|---|---|
| National Health and Medical Research Council | 1025594 | John Silke |
| Australian Cancer Research Foundation | | James M Murphy |
| Thomas William Francis and Violet Coles Trust Fund | | James A Rickard, John Silke |
| State Government of Victoria | Operational Infrastructure Support (OIS) | James A Rickard, Holly Anderton, Nima Etemadi, Ueli Nachbur, Najoua Lalaoui, Kate E Lawlor, Hannah Vanyai, Cathrine Hall, Aleks Bankovacki, Jason Corbin, James M Murphy, Warren S Alexander, Anne K Voss, David L Vaux, John Silke |
| National Institutes of Health | OD012198 | William J Kaiser |
| National Health and Medical Research Council | 1046984 | John Silke |
| National Health and Medical Research Council | 1046010 | John Silke |
| National Health and Medical Research Council | 1051210 | Kate E Lawlor |
| National Health and Medical Research Council | 1057905 | James M Murphy, John Silke |
| National Health and Medical Research Council | 575512 | Anne K Voss |
| National Health and Medical Research Council | 541901 | John Silke |
| National Health and Medical Research Council | 1058190 | John Silke |
| National Health and Medical Research Council | 1058344 | Warren S Alexander |
| National Institutes of Health | GM112547 | Edward S Mocarski |
| Wellcome Trust | 096831/Z/11/Z | Henning Walczak |
| European Research Council | 294880 | Henning Walczak |
| National Health and Medical Research Council | 361646 | James A Rickard, Holly Anderton, Nima Etemadi, Ueli Nachbur, Najoua Lalaoui, Kate E Lawlor, Cathrine Hall, Aleks Bankovacki, Jason Corbin, James M Murphy, Warren S Alexander, Anne K Voss, David L Vaux, John Silke |
| National Health and Medical Research Council | 1016647 | Warren S Alexander |
| National Health and Medical Research Council | 461221 | David L Vaux |

The funders had no role in study design, data collection and interpretation, or the decision to submit the work for publication.

## Author contributions

JAR, Conception and design, Acquisition of data, Analysis and interpretation of data, Drafting or revising the article, Contributed unpublished essential data or reagents; HA, NE, UN, NL, KEL, HV, CH, AB, LG, WW-LW, JC, CH, AKV, Acquisition of data, Analysis and interpretation of data, Drafting or revising the article; MD, NP, Acquisition of data; ESM, HW, Conception and design, Analysis and interpretation of data; JMM, WSA, WJK, Acquisition of data, Analysis and interpretation of data, Drafting or revising the article, Contributed unpublished essential data or reagents; DLV, JS, Conception and design, Analysis and interpretation of data, Drafting or revising the article, Contributed unpublished essential data or reagents

## Ethics

Animal experimentation: Animal experiments were performed in strict accordance with the WEHI Animal Ethics Committee and Institute guidelines. All procedures were specifically approved under WEHI Ethics Project Number 2011.013.

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
