## [Decision Letter]

Thank you for sending your work entitled “TNFR1-dependent cell death drives inflammation in *Sharpin*-deficient mice” for consideration at *eLife*. Your article has been favorably evaluated by Tadatsugu Taniguchi (Senior editor) and three reviewers, one of whom is a member of our Board of Reviewing Editors.

The Reviewing Editor and the other reviewers discussed their comments before we reached this decision, and the Reviewing Editor has assembled the following comments to help you prepare a revised submission.

*Sharpin*^*cpdm/cpdm*^ mice (hereafter referred to as *cpdm* mice) develop severe dermatitis, characterized by extensive apoptotic cell death and multi-organ inflammatory pathologies. SHARPIN, an essential regulator of the LUBAC complex, has been implicated in TNF-induced NF-kB activation and protection against TNF-induced cell death. Previously, TNF and IL1 were both implicated in the development of the pathologies observed in *cpdm* mice. In the current manuscript, the authors wish to address the role of TNF receptors and cell death in the development of the inflammatory phenotypes in *cpdm* mice. The authors show that TNFR1 but not TNFR2 deficiency suppresses the *cpdm* phenotype. IL-R1 deficiency significantly delays the appearance of skin lesions, but eventually these mice develop disease. To investigate the role of cell death in the development of *cpdm*-related pathologies, the authors crossed these mice with caspase-8-, RIPK3-, BID-, and MLKL-deficient mice (and some combinations thereof). Caspase-8^+/–^ prevented the development of skin lesions, but did not rescue spleen and liver inflammation. RIPK3 or MLKL ablation delayed the *cpdm* dermatitis occurrence, but it could not rescue the skin phenotype. In addition, these mice crosses showed reduced spleen inflammation and did not develop liver inflammation at 12 weeks of age. BID deficiency did not affect the *cpdm* phenotype. The fact that cpdm/caspase-8^+/–^ mice did not develop dermatitis, but do develop pathology in the other organs affected in *cpdm* mice, shows the importance of cell death of driving the skin inflammatory phenotype observed in the *cpdm* mice. Combined RIPK3^–/–^ and Casp8^+/–^ completely protected against skin, liver, and spleen pathology in most mice. Surprisingly, triple SHARPIN-, RIPK3- and caspase-8-deficient mice display perinatal lethality. This is remarkable since it is known that RIPK3 ablation rescues early lethality of caspase-8^–/–^ mice. One major issue with the manuscript is that it lacks the mechanistic exploration. The authors should address the following points before a revised manuscript can be considered.

Major comments:

1) The authors have observed keratinocyte cell death in vivo in *Sharpin*-deficient *cpdm* mice. They also show that increased caspase-3/8 cleavage in cultured primary keratinocyte from the *cpdm* mice. A potential important conclusion here is that *Sharpin* negatively regulates caspase-8-mediated apoptosis. The authors should expand this part of work to strengthen the mechanism underlying *Sharpin*-deficiency-induced skin inflammation in *cpdm* mice. They can quantify the cell death percentage by flow cytometry or ATP release assay. Furthermore, they can also analyze RIPK1/3-containing complexes upon TNF stimulation in the presence and absence of *Sharpin*. There are some keratinocyte cell lines such as HaCaT that can be used to perform siRNA knockdown.

2) Sharpin is a component of LUBAC complex. The current manuscript fails to address whether TNF-driven cell death in the absence of *Sharpin* is due to the loss of linear ubiquitin chain synthesis. What is the role of NF-kB signaling in this process? If we knockdown other LUBAC components in keratinocyte, will the cells become more sensitive to TNF-induced cell death? Another possible way to do this is to employ the linear ubiquitin chain DUB Otulin. For example, is there any increased apoptosis sensitivity to TNF stimulation when Otulin is overexpressed in keratinocyte to eliminate the linear ubiquitin chains? If so, is this dependent upon the presence of *Sharpin*?

3) The authors made an interesting and unexpected observation that *Shpn*^m/m^*Casp8*^–/–^Ripk3^–/–^ mice die perinatally. This is stimulating and of potential significance. The authors should better define what is going wrong in the triple knockout mice that die perinatally. Is cell death in certain tissues at the basis of this?

4) Throughout the manuscript, the authors only examine 3–5 mice in many of their experiments. This sample size appears to be too small; for example, in Figures 4 and 2, there are no more than 2 *Shpn*^m/m^*Casp8*^+/–^ mice used to measure epidermal thickness and spleen weight. The authors should increase the number of mice assayed so that the data are more complete and the conclusion is better justified.

5) Figure 2: There seems to be a significant number of anti-active caspase-3 positive cells in the dermis of *cpdm* mice at 9 weeks of age that do not seem to be associated with hair follicles. Are these apoptotic dermal fibroblast or another cell type (perhaps immune cells)? Characterizing these cells could be important in relation to the results presented in Figure 5–figure supplement 1.

6) The data presented in Figure 3 and Figure 3–figure supplement 1 point out that the role of *Sharpin* in TNF-induced NF-kB activation could be different in keratinocytes compared to other cell types. One cannot observe a convincing difference in IkBα phosphorylation or degradation in these cells upon TNF stimulation. It would be interesting to compare the effects of *Sharpin* deficiency on TNF-induced NF-kB activation in MEFs, keratinocytes, and dermal fibroblasts. It may be that in certain cell types the main effect of *Sharpin* depletion is at the level of regulating cell death rather than NF-kB activation. This could also be stressed in the Discussion.

7) Figure 5–figure supplement 1: Since TNF can induce, depending on the conditions, RIPK1-dependent apoptosis and necroptosis, and since cell death modes can switch from apoptosis to necrosis, and vice versa, when using apoptosis or necroptosis inhibitors, we cannot conclude which primary cell death mode is operating in *cpdm* dermal fibroblast without showing analysis of caspase activation in the different conditions used (western blot or DEVD assays).

---

## [Author Response]

*1) The authors have observed keratinocyte cell death* in vivo *in Sharpin-deficient cpdm mice. They also show that increased caspase-3/8 cleavage in cultured primary keratinocyte from the cpdm mice. A potential important conclusion here is that Sharpin negatively regulates caspase-8-mediated apoptosis. The authors should expand this part of work to strengthen the mechanism underlying Sharpin-deficiency-induced skin inflammation in cpdm mice. They can quantify the cell death percentage by flow cytometry or ATP release assay. Furthermore, they can also analyze RIPK1/3-containing complexes upon TNF stimulation in the presence and absence of Sharpin. There are some keratinocyte cell lines such as HaCaT that can be used to perform siRNA knockdown*.

We have added two separate pieces of data that address these points. In our new Figure 4 – we show that primary *Shpn*^*m/m*^ keratinocytes are sensitive to TNF-induced cell death and that this is blockable by both Q-VD-OPh and Nec-1 with the combination providing the best protection. This is entirely consistent with what we previously showed in Gerlach et al, using a different assay and with the results from Kumari et al (20-05-2014-RA-eLife-03422R2). In our new Figure 4 (and new Figure 4—figure supplement 1) we show that primary *Shpn*^*m/m*^ dermal fibroblasts rapidly activate caspase-8 and caspase-3 in response to TNF. Remarkably, Nec-1 prevents this activation. This data strongly supports our previous conclusion that the kinase activity of RIPK1 is required to activate caspases in the skin. We have not succeeded in analysing RIPK1/RIPK3 complexes in these primary cell types due to the limited amount of material.

*2) Sharpin is a component of LUBAC complex. The current manuscript fails to address whether TNF-driven cell death in the absence of Sharpin is due to the loss of linear ubiquitin chain synthesis. What is the role of NF-kB signaling in this process? If we knockdown other LUBAC component in keratinocyte, will the cells become more sensitive to TNF-induced cell death? Another possible way to do this is to employ the linear ubiquitin chain DUB Otulin. For example, is there any increased apoptosis sensitivity to TNF stimulation when Otulin is overexpressed in keratinocyte to eliminate the linear ubiquitin chains? If so, is this dependent upon the presence of Sharpin*?

We have now shown that SHARPIN deficiency leads to a reduction in linear ubiquitylation of components of the TNFR1 signaling complex in response to TNF (new Figure 5). In *Shpn*^*m/m*^ primary keratinocytes and dermal fibroblasts NF-κB activation in response to TNF is not dramatically altered compared with wild-type cells in the 15 min – 4 hr window (Figure 4 and new Figure 4). Furthermore, cFLIP cleavage occurs within 15 minutes of TNF treatment in dermal fibroblasts and detectable caspase processing occurs soon after. Therefore, in this instance, it seems unlikely that a reduction in NF-κB is responsible for TNF-induced cell death (Figure 4 and new Figure 4—figure supplement 1). We have also shown that HaCaT keratinocyte cells stably expressing the catalytically inactive HOIP^C885S^ mutant, but not wild-type HOIP, are sensitive to TNF-induced cell death (Figure 5). With regard to other LUBAC components, our results in *Shpn*^*m/m*^ cells are consistent with those recently described for HOIP-deficient cells in Peltzer et al (36). These cells have lost the ability to induce linear ubiquitin chains in response to TNF and were sensitive to TNF-induced cell death that was reduced by Q-VD-OPh and Nec-1. We have now cited this work in our manuscript.

*3) The authors made an interesting and unexpected observation that Shpn*^*m/m*^*Casp8*^*–/–*^*Ripk3*^*–/–*^
*mice die perinatally. This is stimulating and of potential significance. They should better define what is going wrong in the triple knockout mice that die perinatally. Is cell death in certain tissues at the basis of this*?

In response to the reviewers' comments we approached the lab of Henning Walczak for help with TNFR1 complex analysis. In the course of this conversation we learned that they had also generated *Shpn*^*m/m*^*Casp8*^*–/–*^*Ripk3*^*–/–*^ mice. As described in our first submission, we have not generated any mice that survived until weaning and many were already dead when we explored the timing of death using E19 caesareans in either the Silke (Australia) or the Kaiser facility (Atlanta, USA). The table in Figure 8 represents only data from the Silke lab. We have included a summary of the results from the Kaiser facility below for the reviewers.

Lethality of *Shpn*^*m/m*^*Casp8*^*–/–*^*Ripk3*^*–/–*^ from Kaiser facility in Emory University.

Somewhat remarkably to us, the Walczak lab (London, UK) obtained two mice that survived past weaning. Because the source of the caspase-8 knockout strain is different between the Silke lab (Stephen Hedrick; exon 3) and the Kaiser and Walczak lab (Rasqallah Hakem; exons 3 & 4) it is possible that there is some genetic difference that accounts for some of the difference in lethality – note that while most of the Kaiser triple knockout mice die at day 0–1 there are a couple of outliers at 12–16, whereas in the Silke lab we have never observed anything past day 1. It is also clear, however, that the environment plays an important role in the lethality, because the Kaiser lab never observed any triple knockout mice that survived past weaning. Unfortunately, however, this spread in the timing of the lethality makes it very difficult to address the cause. We have highlighted this issue in our new text and analysed the living, runted, mutant mice, which we believe makes the spread in the lethality abundantly obvious. This data is presented in Figure 8. Consistent with the *Shpn*^*m/m*^*Casp8*^+/–^ phenotype (Figure 6), the epidermal hyperplasia is absent in the *Shpn*^*m/m*^*Casp8*^*–/–*^*Ripk3*^*–/–*^ triple mutant mice. Consistent with the *Shpn*^*m/m*^*Ripk3*^*–/–*^ phenotype (Figure 7), the *Shpn*^*m/m*^*Casp8*^*–/–*^*Ripk3*^*–/–*^ triple mutant mice do not have dramatic splenomegaly.

*4) Throughout the manuscript, the authors only examine 3–5 mice in many of their experiments. This sample size appears to be too small; for example, in*
Figures 4 and 2*, there are no more than 2 Shpn*^*m/m*^*Casp8*^*+/–*^
*mice used to measure epidermal thickness and spleen weight. The authors should increase the number of mice assayed so that the data are more complete and the conclusion is better justified*.

Throughout the paper we have increased numbers where possible, especially for analysis of the *Shpn*^*m/m*^*Casp8*^–/+^ (epidermis n=8) and *cpdmMlkl*^*–/–*^ (n=26) crosses. Unfortunately during the review process our *Shpn*^*m/m*^*Casp8*^–/+^ line was affected by *Pasteurella*, and most of the target genotype required early euthanasia due to a pulmonary infection. This would appear to be due to a compromised immune system in these mice because neither the littermates, nor any other strain in the facility, were affected. We measured skin thickness because this is unlikely to be affected by a pulmonary infection, and indeed our measurements support this notion. These mice, however, were inappropriate for analysis of spleen and blood. We analysed spleen and blood of two additional mice that did not show signs of infection so n is now equal to three. Although these two additional mice did not show signs of infection, we cannot exclude that there is an occult infection. That being said, the spleen weight values for the three *Shpn*^*m/m*^*Casp8*^–/+^ are tightly distributed. For accuracy and transparency purposes we have described this complication in detail in the results section of the manuscript. Unfortunately there is little that we can do to increase numbers further; breeding more mice will take months and there remains the possibility that these mice will again succumb to this opportunistic infection.

*5)*
Figure 2*: There seems to be a significant number of anti-active caspase-3-positive cells in the dermis of cpdm mice at 9 weeks of age that do not seem to be associated with hair follicles. Are these apoptotic dermal fibroblast or another cell type (perhaps immune cells)? Characterizing these cells could be important in relation to the results presented in Figure 5–figure supplement 1*.

Since the *cpdm* phenotype can occur in the absence of lymphocytes in RAG knockout mice (37) and macrophages are elevated in *cpdm* skin (Figure 2), we tested the sensitivity of *Shpn*^*m/m*^ macrophages, neutrophils and monocytes to TNF (new Figure 4). *Shpn*^*m/m*^ neutrophils are sensitive to TNF-induced cell death and this death is blockable by Q-VD-OPh and Nec-1. Similarly to *Shpn*^*m/m*^ dermal fibroblasts, *Shpn*^*m/m*^ macrophages are also sensitive to TNF-induced cell death and this death is almost completely inhibited by Nec-1, but far less so by Q-VD-OPh (cf Figure 4). *Shpn*^*m/m*^ monocytes are intrinsically less viable than wild-type monocytes; this intrinsic reduction in viability is reduced by Q-VD-OPh but not by Nec-1. Like the other cell types, these *Shpn*^*m/m*^ monocytes are more sensitive to TNF-induced cell death than the wild-type cells. These results show that the major cell types present in the dermis and epidermis of *Shpn*^*m/m*^ mutant mice (keratinocytes, dermal fibroblasts, and macrophages) are all sensitive to TNF-induced death that is blocked by Nec-1. It is, therefore, possible that cell death of any of these cells contributes to the epidermal hyperplasia.

*6) The data presented in*
Figure 3
*and Figure 3–figure supplement 1 point out that the role of Sharpin in TNF-induced NF-kB activation could be different in keratinocytes compared to other cell types. One cannot observe a convincing difference in IkBα phosphorylation or degradation in these cells upon TNF stimulation. It would be interesting to compare the effects of Sharpin deficiency on TNF-induced NF-kB activation in MEFs, keratinocytes, and dermal fibroblasts. It may be that in certain cell types the main effect of sharpin depletion is at the level of regulating cell death rather than NF-kB activation. This could also be stressed in the Discussion*.

As requested, we have now performed a signaling analysis in dermal fibroblasts (new Figure 4). Similarly to *Shpn*^*m/m*^ keratinocytes, we observe that *Shpn*^*m/m*^ dermal fibroblasts are able to activate NF-κB in response to TNF in a very similar fashion to wild-type fibroblasts, as judged by the kinetics of IκBα degradation and p65, JNK and p38 phosphorylation (new Figure 4). cFLIP processing, however, occurs very rapidly within 15 min of TNF addition. Given this timing and the apparently normal NF-κB response, it seems unlikely that loss of NF-κB is the cause of caspase activation in these *Shpn*^*m/m*^ cells. Therefore, we agree with the reviewer that this indicates that, in these cell types, SHARPIN is doing something to prevent TNF-induced cell death that is not dependent on NF-κB activation. This is an interesting observation, because in previous studies looking at *Shpn*^*m/m*^ or *Hoip*^–/–^ MEFs, they were defective both in NF-κB and sensitive to TNF, making it impossible to differentiate the two different effects (15; 23; 49; 36). This is, therefore, an important result that we have highlighted in the Discussion section.

*7) Figure 5–figure supplement 1: Since TNF can induce, depending on the conditions, RIPK1-dependent apoptosis and necroptosis, and since cell death modes can switch from apoptosis to necrosis, and vice versa, when using apoptosis or necroptosis inhibitors, we cannot conclude which primary cell death mode is operating in cpdm dermal fibroblast without showing analysis of caspase activation in the different conditions used (western blot or DEVD assays)*.

As requested, we have western blotted for cFLIP and caspases in purified dermal fibroblasts treated with TNF (new Figure 4) obtaining the same results that we saw in keratinocytes (Figure 4). The cleavage of cFLIP and the activation of caspase-8 and caspase-3 are all consistent with an apoptotic death. These results are also entirely consistent with the fact that *Casp8* heterozygosity, but not loss of *Mlkl,* prevents hyperplasia and the appearance of cleaved caspase-3 in *Shpn*^*m/m*^ skin (Figures 6 and 7). Most remarkably however, and as noted before, Nec-1 prevents the activation of caspase-3 and -8. This beautifully, albeit unintentionally on our part, supports the point that the reviewers make, namely that it is very dangerous to conclude the mode of cell death based upon inhibitors only. And it shines an interesting light on the recent publication from Berger et al, which shows that the kinase-dead RIPK1 knock-in mouse prevents the *Shpn*^*m/m*^ phenotype from being observed (4). Again, this is appropriately discussed in our revised manuscript.